# Certifiably Robust Graph Contrastive Learning

**Minhua Lin, Teng Xiao, Enyan Dai, Xiang Zhang, Suhang Wang**
The Pennsylvania State University
`{mfl5681,tengxiao,emd5759,xzhang,szw494}@psu.edu`

## Abstract

Graph Contrastive Learning (GCL) has emerged as a popular unsupervised graph representation learning method. However, it has been shown that GCL is vulnerable to adversarial attacks on both the graph structure and node attributes. Although empirical approaches have been proposed to enhance the robustness of GCL, the certifiable robustness of GCL is still remain unexplored. In this paper, we develop the first certifiably robust framework in GCL. Specifically, we first propose a unified criteria to evaluate and certify the robustness of GCL. We then introduce a novel technique, **RES** (Randomized Edgedrop Smoothing), to ensure certifiable robustness for any GCL model, and this certified robustness can be provably preserved in downstream tasks. Furthermore, an effective training method is proposed for robust GCL. Extensive experiments on real-world datasets demonstrate the effectiveness of our proposed method in providing effective certifiable robustness and enhancing the robustness of any GCL model. The source code of RES is available at `https://github.com/ventr1c/RES-GCL`.

## 1 Introduction

Graph structured data are ubiquitous in real-world applications such as social networks [1], finance systems [2], and molecular graphs [3]. Graph Neural Networks (GNNs) have emerged as a popular approach to learn graph representations by adopting a message passing scheme [4, 5, 6], which updates a node's representation by aggregating information from its neighbors. Recently, graph contrastive learning has gained popularity as an unsupervised approach to learn node or graph representations [7, 8, 9]. The graph contrastive learning creates augmented views and minimizes the distances among positive pairs while maximizing the distances among negative pairs in the embedding space.

Despite the great success of GNNs, some existing works [10, 11, 12, 13] have shown that GNNs are vulnerable to adversarial attacks where the attackers can deliberately manipulate the graph structures and/or node features to degrade the model's performance. The representations learned by Graph Contrastive Learning (GCL) are also susceptible to such attacks, which can lead to poor performance in downstream tasks with a small amount of perturbations [14, 15, 16]. To defend against graph adversarial attacks, several empirical-based works have been conducted in robust GNNs for classification tasks [17, 18] and representation learning with contrastive learning [14, 15]. Nevertheless, with the advancement of new defense strategies, new attacks may be developed to invalidate the defense methods [19, 20, 21, 22], leading to an endless arms race. Therefore, it is important to develop a certifiably robust GNNs under contrastive learning, which can provide certificates to nodes/graphs that are robust to potential perturbations in considered space. There are several efforts in certifiable robustness of GNNs under supervised learning, which requires labels for the certifiable robustness analysis [23, 24, 25, 26]. For instance, [23] firstly analyze the certifiable robustness against the perturbation on node features. Some following works [24, 25, 26] further study the certified robustness of GNN under graph topology attacks. However, the certifiable robustness of GCL, which is in unsupervised setting, is still rarely explored.

37th Conference on Neural Information Processing Systems (NeurIPS 2023).

Certifying the robustness of GCL is rather challenging. *First*, quantifying the robustness of GCL consistently and reliably is a difficult task. Existing GCL methods [27, 14, 15] often use empirical robustness metrics (e.g., robust accuracy) that rely on the knowledge of specific downstream tasks (e.g., task types, node/graph labels) for evaluating GCL robustness. However, these criteria may not be universally applicable across different downstream tasks. The lack of clarity in defining and assessing robustness within GCL further compounds the difficulty of developing certifiably robust GCL methods. *Second*, the absence of labels in GCL makes it challenging to study certifiable robustness. Existing works [23, 28, 29, 25, 24] on certifiable robustness of GNNs are designed for supervised or semi-supervised settings, which are not applicable to GCL. They typically try to provide certificates by analyzing the worst-case attack on labeled nodes. However, it is difficult to conduct such analyses due to the absence of labels in GCL. Additionally, some works [29, 24] that rely on randomized smoothing to obtain robustness certificates can not be directly applied to GCL as they may require injecting noise into the whole graph during data augmentation, introducing too many spurious edges and hurting downstream task performance, particularly for large and sparse graphs. *Third*, it is unclear whether certifiable robustness of GCL can be converted into robust performance in downstream tasks. The challenge arises due to the misalignment of objectives between GCL and downstream tasks. For instance, GCL usually focuses on discriminating between different instances, while some downstream tasks may also require understanding the relations among instances. This discrepancy engenders a divide between unsupervised GCL and its downstream tasks, making it difficult to prove the robust performance of the certified robust GNN encoder in downstream tasks.

In this work, we propose the first certifiably robust GCL framework, RES (Randomized Edgedrop Smoothing). **(i)** To address the ambiguity in evaluating and certifying robustness of GCL, we propose a unified criteria based on the semantic similarity between node or graph representations in the latent space. **(ii)** To address the label absence challenges and avoid adding too many noisy edges for certifying the robustness in GCL, we employ the robustness defined at (i) for unlabeled data and introduce a novel technique called randomized edgedrop smoothing, which can transform any base GNN encoder trained by GCL into a certifiably robust encoder. Therein, some randomized edgedrop noises are injected into the graphs by stochastically dropping observed edges with a certain probability, preventing the introduction of excessive noisy edges and yielding effective and efficient performance. **(iii)** We theoretically demonstrate that the representations learned by our robust encoder can achieve provably robust performance in downstream tasks. **(iv)** Moreover, we present an effective training method that enhances the robustness of GCL by incorporating randomized edgedrop noise. Our experimental results show that GRACE [30] with our RES can outperform the state-of-the-art baselines against several structural attacks and also achieve 43.96% certified accuracy in OGB-arxiv dataset when the attacker arbitrarily adds at most 10 edges.

Our **main contributions** are: **(i)** We study a novel problem of certifying the robustness of GCL to various perturbations. We introduce a unified definition of robustness in GCL and design a novel framework RES to certify the robustness; **(ii)** Theoretical analysis demonstrates that the representations learned by our robust encoder can achieve provably robust performance in downstream tasks; **(iii)** We design an effective training method for robust GCL by incorporating randomize edgedrop noise; and **(iv)** Extensive empirical evaluations show that our method can provide certifiable robustness for downstream tasks (i.e., node and graph classifications) on real-world datasets, and also enhance the robust performance of any GNN encoder through GCL.

## 2 Related Work

**Graph Contrastive Learning.** GCL has recently gained significant attention and shows promise for improving graph representations in scenarios where labeled data is scarce [31, 32, 9, 30, 27, 15, 33, 32, 34, 35]. Generally, GCL creates two views through data augmentation and contrasts representations between two views. Several recent works [16, 36] show that GCL is vulnerable to adversarial attacks. Despite very few empirical works [9, 15, 14] on robustness of GCL, there are no existing works studying the certified robustness of GCL. In contrast, we propose to provide robustness certificates for GCL by using randomized edgedrop smoothing. More details are shown in Appendix A.1. To the best of our knowledge, our method is the first work to study the certified robustness of GCL.

**Certifiable Robustness of GNNs.** Several recent studies investigate the certified robustness of GNNs in the (semi)-supervised setting[23, 28, 29, 26, 24, 25, 37]. Zunger et al. [23] are the first to explore

certifiable robustness with respect to node feature perturbations. Subsequent works[28, 26, 29, 24, 38] extend the analysis to certifiable robustness under topological attacks. More details are shown in Appendix A.2. Our work is inherently different from them: (i) we provide an unified definition to evalute and certify the robustness of GCL; (ii) we theoretically provide the certified robustness for GCL in the absence of labeled data, and this certified robustness can provably sustained in downstream tasks; (iii) we design an effective training method to enhance the robustness of GCL.

## 3 Backgrounds and Preliminaries

### 3.1 Graph Contrastive Learning

**Notations.** Let $\mathcal{G} = (\mathcal{V}, \mathcal{E}, \mathbf{X})$ denote a graph, where $\mathcal{V} = \{v_1, \ldots, v_N\}$ is the set of $N$ nodes, $\mathcal{E} \subseteq \mathcal{V} \times \mathcal{V}$ is the set of edges, and $\mathbf{X} = \{\mathbf{x}_1, ..., \mathbf{x}_N\}$ is the set of node attributes with $\mathbf{x}_i$ being the node attribute of $v_i$. $\mathbf{A} \in \mathbb{R}^{N \times N}$ is the adjacency matrix of $\mathcal{G}$, where $\mathbf{A}_{ij} = 1$ if nodes $v_i$ and $v_j$ are connected; otherwise $\mathbf{A}_{ij} = 0$. In this paper, we focus on unsupervised graph contrastive learning, where label information of nodes and graphs are unavailable during training for both node- and graph-level tasks. For the node-level task, given a graph $\mathcal{G}$, the goal is to learn a GNN encoder $h$ to produce representation $\mathbf{z}_v$ for $v$, which can be used to conduct prediction for node $v$ in downstream tasks. For the graph-level task, given a set of graph $\mathbb{G} = \{\mathcal{G}_1, \mathcal{G}_2, \cdots\}$, the goal is to learn the latent representation $\mathbf{z}_\mathcal{G}$ for each graph $\mathcal{G}$, which can be used to predict downstream label $y_\mathcal{G}$ of $\mathcal{G}$. For simplicity and clarity, in this paper, we uniformly denote a node or graph as a concatenation vector $\mathbf{v}$. The representation of $\mathbf{v}$ from encoder $h$ is denoted as $h(\mathbf{v})$. More details are shown in Appendix E.2.

**Graph Contrastive Learning.** Generally, GCL consists of three steps: (i) multiple views are generated for each instance through stochastic data augmentation. Positive pairs are defined as two views generated from the same instance; while negative pairs are sampled from different instances; (ii) these views are fed into a set of GNN encoders $\{h_1, h_2, \cdots\}$, which may share weights; (iii) a contrastive loss is applied to minimize the distance between positive pairs and maximize the distance between negative pairs in latent space. To achieve the certified robustness of GCL to downstream tasks, we introduce *latent class* to formalize the contrastive loss for GCL, which is inspired by [39, 40].

**Definition 1** (Latent Class). We utilize the latent class to formalize semantic similarity of positive and negative pairs in GCL [39]. Consider a GNN encoder $h$ learnt by GCL. Let $\mathcal{V}$ denote the set of all possible nodes/graphs in the input space. There exists a set of *latent class*, denoted as $\mathcal{C}$, where each sample $\mathbf{v} \in \mathcal{V}$ is associated with a latent class $c \in \mathcal{C}$. Each class $c \in \mathcal{C}$ is associated with a distribution $\mathcal{D}_c$ over the latent space of samples belonging to class $c$ under $h$. The distribution on $\mathcal{C}$ is denoted as $\eta$. Intuitively, $\mathcal{D}_c(\mathbf{v})$ captures how relevant $\mathbf{v}$ is to class $c$ in the latent space under $h$, similar to the meaning of class in supervised settings. The latent class is related to the specific downstream task. For instance, when the downstream task is a classification problem, the latent class can be interpreted as the specific class to which the instances belong.

Typically, GCL is based on the label-invariant augmentation intuition: the augmented operations preserve the nature of graphs and make the augmented positive views have consistent latent classes with the original ones [41]. Let $c^+, c^- \in \mathcal{C}$ denote the positive and negative latent classes drawn from $\eta$, $\mathcal{D}_{c^+}$ and $\mathcal{D}_{c^-}$ are the distributions to sample positive and negative samples, respectively. For each positive pair $(\mathbf{v}, \mathbf{v}^+) \sim \mathcal{D}_{c^+}^2$ associated with $n$ negative samples $\{\mathbf{v}_i^- \sim \mathcal{D}_{c^-} | i \in [n]\}$, the widely-used loss for GCL, which is also known as InfoNCE loss [42], can be written as follows:

$$\mathcal{L}_{GCL} = \mathop{\mathbb{E}}_{c^+, c^- \sim \eta^2} \Big[ \mathop{\mathbb{E}}_{\mathbf{v}, \mathbf{v}^+ \sim \mathcal{D}_{c^+}^2, \mathbf{v}_i^- \sim \mathcal{D}_{c^-}} [-\log(\frac{e^{h(\mathbf{v})^\top h(\mathbf{v}^+)}}{e^{h(\mathbf{v})^\top h(\mathbf{v}^+)} + \sum_{i=1}^{n} e^{h(\mathbf{v})^\top h(\mathbf{v}_i^-)}})] \Big], \tag{1}$$

### 3.2 Threat Model

**Attacker's Goal.** We consider an attacker conducts an evasion attack in GCL. Given a well-trained GNN encoder $h$ on a clean graph $\mathcal{G}$ via GCL, an attacker aims to degrade the performance of $h$ in downstream tasks by injecting noise into the graph. For instance, an attacker may attempt to manipulate a social network by injecting fake edges, which could affect the performance of a well-trained $h$ in tasks such as community detection, node classification, or link prediction. The noise can take different forms, including adding/deleting nodes/edges or augmenting node features.

**Attacker's Knowledge and Capability.** We assume that the attacker follows the grey-box setting to conduct an evasion attack. Specifically, the attacker has access to a benign GNN encoder $h$ trained on a clean dataset through GCL, and the training data to train downstream classifier is available to the attacker. The model architecture and other related information of the GNN encoder are unknown to the attacker. During inference phase, the attacker is capable of injecting noises to the graph within a given budget to degrade the performance of target nodes/graphs in downstream tasks. In this paper, we focus on perturbations on the graph structure $\mathbf{A}$, i.e, only structural noises such as adding new edges/nodes are injected into the graph as structure attack is more effective than feature attack [11, 24, 43]. We leave the extension to feature attack and defense of GCL as future work. We denote $\delta \in \{0,1\}^N$ as the structural noise to a node/graph $\mathbf{v}$, where $N$ denote the number of nodes on the graph $\mathbf{v}$ or $K$-hop subgraph of node $\mathbf{v}$ and $\delta_i = 1$ indicates the addition of a noisy edge to $\mathbf{v}$ in the $i$-th entry, and its $L_0$-norm $||\delta||_0$ indicates the number of noisy edges. We then have the perturbed version of $\mathbf{v}$ by the attacker, denoted as $\mathbf{v}' = \mathbf{v} \oplus \delta$.

### 3.3 Problem Statement

Our objective is to develop a certifiably robust GCL. We aim to train a GNN encoder that exhibits provably benign behaviors in downstream tasks. The problem can be formulated as follows:

**Problem 1.** *Given a GNN encoder $h$, a node or graph $\mathbf{v}$. Let $\mathbf{v}' = \mathbf{v} \oplus \delta$ denote the perturbed version of $\mathbf{v}$ by the attacker, where the structural noise $\delta$ with $||\delta||_0 \leq k$ is injected into $\mathbf{v}$. Suppose that $c^*$ is the latent class of $\mathbf{v}$ and $f$ is the downstream classifier. Our goal is to develop a certifiably robust GCL for $h$ such that for any $c \neq c^*$, the representation of $\mathbf{v}'$ under $h$ can satisfy the following requirement in downstream tasks:*

$$m(\mathbf{v}', c^*; h) = \min_{c \neq c^*} \left[ \mathbb{P}(f(h(\mathbf{v}')) = c^*) - \mathbb{P}(f(h(\mathbf{v}')) = c) \right] > 0, \tag{2}$$

*where $m(\mathbf{v}', c^*; h)$ is the worst-case margin of $h(\mathbf{v}')$ between $c^*$ and $y$ in the downstream task for any $c \neq c^*$, and $\mathbb{P}(f(h(\mathbf{v}')) = c)$ denotes the probability of the representation $h(\mathbf{v}')$ being classified into class $c$ by $f$ in the downstream task. Eq. (2) implies that $f(h(\mathbf{v}')) = f(h(\mathbf{v})) = c^*$ for any $\delta$ within the budget $k$, i.e., $h$ is a certifiably robust at $(\mathbf{v}, c^*)$ when perturbing at most $k$ edges.*

## 4 Certifying Robustness for GCL

In this section, we present the details of our RES, which aims to provide certifiable robustness for GCL models. There are mainly three challenges: (i) how to define the certified robustness of GCL; (ii) how to derive the certified robustness; and (iii) how to transfer the certified robustness of GCL to downstream tasks. To address these challenges, we first define certified robustness for GCL in Sec. 4.1. We then propose the RES method to derive the certificates and theoretically guarantee certifiable robustness in Sec. 4.2. Finally, we show that the certifiably robust representations learned from our approach are still provably robust in downstream tasks in Sec. 4.3.

### 4.1 Certified Robustness of GCL

To give the definition of certified robustness of GCL, we first give the conditions for successfully attacking the GNN encoder. The core idea behind it comes from supervised learning, where the objective is to judge whether the predictions of target nodes/graphs are altered under specific perturbations. Inspired by [40], we consider the following scenario of a successful attack against the GNN encoder.

Given a clean input $\mathbf{v}$, with positive and negative samples $\mathbf{v}^+$ and $\mathbf{v}^-$ used in the learning process of GCL, respectively, we consider $\mathbf{v}' = \mathbf{v} \oplus \delta$, where $\delta$ represents structural noise on $\mathbf{v}$ and $\mathbf{v}^-$ is the attack target of $\mathbf{v}$. The attacker's goal is to produce an adversarial example $\mathbf{v}'$ that can deceive the model into classifying $\mathbf{v}$ as similar to $\mathbf{v}^-$. Formally, given a well-trained GNN encoder $h$ via GCL, we say that $h$ has been successfully attacked at $\mathbf{v}$ if the cosine similarity $s\left(h(\mathbf{v}'), h(\mathbf{v}^+)\right)$ is less than $s\left(h(\mathbf{v}'), h(\mathbf{v}^-)\right)$. This indicates that $\mathbf{v}'$ is more similar to $\mathbf{v}^-$ than to $\mathbf{v}^+$ in the latent space. Otherwise, we conclude that $h$ has not been successfully attacked. This definition of successful attack provides the basis for the definition of certified robustness of GCL. The formal definition of certified robustness problem for GCL is then given as

**Definition 2** (Certified Robustness of GCL)**.** Given a well trained GNN encoder $h$ via GCL. Let $\mathbf{v}$ be a clean input. $\mathbf{v}^+$ is the positive sample of $\mathbf{v}$ and $\mathbf{V}^- = \{\mathbf{v}_1^-, \cdots, \mathbf{v}_n^-\}$ denotes all possible negative

samples of $\mathbf{v}^+$, which are sampled as discussed in Sec. 3.1. Suppose that $\mathbf{v}'$ is a perturbed sample obtained by adding structural noise $\delta$ to $\mathbf{v}$, where $||\delta||_0 \leq k$, and $s(\cdot, \cdot)$ is a cosine similarity function. Then, $h$ is certifiably $l_0^k$-robust at $(\mathbf{v}, \mathbf{v}^+)$ if the following inequality is satisfied:

$$s(h(\mathbf{v}'), h(\mathbf{v}^+)) > \max_{\mathbf{v}^- \in \mathbf{V}^-} s(h(\mathbf{v}^-), h(\mathbf{v}')), \ \forall \delta : \|\delta\|_0 \leq k. \tag{3}$$

Similar to the supervised certified robustness, this problem is to prove that the point $\mathbf{v}' = \mathbf{v} \oplus \delta$ under the perturbation within budget $k$ is still the positive sample of $\mathbf{v}^+$. However, unlike the supervised learning, where we can estimate the prediction distribution of $\mathbf{v}$ by leveraging the labels of the training data, it is difficult to estimate such a distribution in GCL because the label information is unavailable. To address this issue, we first consider a space $\mathbb{B}$ for a sample $\mathbf{v}^+$, where each sample within it is the positive sample of $\mathbf{v}^+$. Formally, given a sample $\mathbf{v}^+$ with the latent class $c^+$ from the probability distribution $\mathcal{D}_c^+$, $\mathbb{B}(\mathbf{v}^+)$ is defined as

$$\mathbb{B}(\mathbf{v}^+) := \{\mathbf{v} | \mathbf{v} \in c^+\}. \tag{4}$$

GCL maximizes the agreement of the positive pair in the latent space and assume access to *similar* data in the form of pairs $(\mathbf{v}, \mathbf{v}^+)$ that comes from a distribution $\mathcal{D}_{c^+}^2$ given the latent class $c^+$. Thus, it is natural to connect the latent class and the similarity between representations. The probability that $\mathbf{v}$ is the positive sample of $\mathbf{v}^+$ can be given by the following theorem.

**Theorem 1.** *Let $\mathbb{B}(\mathbf{v}^+)$ be a space around $\mathbf{v}^+$ as defined in Eq. (4). Given an input sample $\mathbf{v}$ and a GNN encoder $h$ learnt by GCL in Eq. (1), the probability of $\mathbf{v}$ being the positive sample of $\mathbf{v}^+$ is:*

$$Pr(\mathbf{v} \in \mathbb{B}(\mathbf{v}^+); h) = \exp\left[-\left(\frac{1 - s(h(\mathbf{v}), h(\mathbf{v}^+))}{a}\right)^\sigma\right], \tag{5}$$

*where $s(\cdot, \cdot)$ is the cosine similarity function. $a, \sigma > 0$ are Weibull shape and scale parameters [44].*

The proof is in Appendix C.1. Theorem 1 manifests that we can derive the probability of $\mathbf{v}$ being the positive sample of $\mathbf{v}^+$ based on the cosine similarity under $h$, providing us a feasible way to study the certified robustness problem for GCL without the label information.

## 4.2 Certifying Robustness by the Proposed Randomized EdgeDrop Smoothing (RES)

With the definition of certifiable robustness of GCL, we introduce our proposed Randomized Edgedrop Smoothing (RES), which provides certifiable robustness for GCL. RES injects randomized edgedrop noise by randomly dropping each edge of the input sample with a certain probability. We will also demonstrate the theoretical basis for the certifiable robustness of GCL achieved by RES.

Firstly, we define an randomized edgedrop noise $\epsilon$ for $\mathbf{v}$ that remove each edge of $\mathbf{v}$ with the probability $\beta$. Formally, given a sample $\mathbf{v}$, the probability distribution of $\epsilon$ for $\mathbf{v}$ is given as:

$$\mathbb{P}(\epsilon_i = 0 | \mathbf{v}_i = 0) = 1, \ \mathbb{P}(\epsilon_i = 0 | \mathbf{v}_i = 1) = \beta, \ \text{and} \ \mathbb{P}(\epsilon_i = 1 | \mathbf{v}_i = 1) = 1 - \beta, \tag{6}$$

where $\mathbf{v}_i$ denotes the connection status of $i$-th entry of $\mathbf{v}$. Given a well-trained GNN encoder $h$ via GCL and a noise $\epsilon$ with the distribution in Eq. (6), the smoothed GNN encoder $g$ is defined as:

$$g(\mathbf{v}) = h(\mathbf{v} \oplus \epsilon), \ \text{where} \ \mathbf{v} \oplus \epsilon \in \mathbb{B}(\arg\max_{\hat{\mathbf{v}} \in \mathbf{V}} \mathbb{P}(\mathbf{v} \oplus \epsilon \in \mathbb{B}(\hat{\mathbf{v}}); h)). \tag{7}$$

Here $\mathbf{V}$ is the set of all nodes/graphs in the dataset. It implies $g$ will return the representation of $\mathbf{v} \oplus \delta$ whose latent class is the same as that of the most probable instance that $\mathbf{v} \oplus \epsilon$ is its positive sample. Therefore, consider the scenario where some noisy edges are injected into a clean input $\mathbf{v}$, resulting in a perturbed sample $\mathbf{v}'$. With a smoothed GNN encoder $g$ learnt through our method, if we set $\beta$ to a large value, it is highly probable that the noisy edges will be eliminated from $\mathbf{v}'$. Consequently, by running multiple randomized edgedrop on $\mathbf{v}$ and $\mathbf{v}'$, a majority of them will possess identical structural vectors, that is, $\mathbf{v} \oplus \epsilon = \mathbf{v}' \oplus \epsilon$, which implies $\mathbf{v}$ and $\mathbf{v}'$ will have the same latent class and the robust performance of $\mathbf{v}'$ in downstream tasks. We can then derive an upper bound on the expected difference in vote count for each class between $\mathbf{v}$ and $\mathbf{v}'$. With this, we can theoretically demonstrate that $g$ is certifiably robust at $\mathbf{v}$ against any perturbation within a specific attack budget. Specifically, let $p_{\mathbf{v}^+, h}(\mathbf{v}) = \mathbb{P}(\mathbf{v} \in \mathbb{B}(\mathbf{v}^+); h)$ for simplicity of notation, $\underline{p_{\mathbf{v}^+, h}}(\mathbf{v} \oplus \epsilon)$ denotes a lower bound on $p_{\mathbf{v}^+, h}(\mathbf{v} \oplus \epsilon)$ with $(1 - \alpha)$ confidence and $\overline{p_{\mathbf{v}_i^-, h}}(\mathbf{v} \oplus \epsilon)$ denotes a similar upper bound, the formal theorem is presented as follows:

**Theorem 2.** *Let $\mathbf{v}$ be a clean input and $\mathbf{v}' = \mathbf{v} \oplus \delta$ be its perturbed version, where $||\delta||_0 \leq k$. $\mathbf{V}^- = \{\mathbf{v}_1^-, \cdots, \mathbf{v}_n^-\}$ is the set of negative samples of $\mathbf{v}$. If for all $\mathbf{v}_i^- \in \mathbf{V}^-$:*

$$\underline{p_{\mathbf{v}^+, h}}(\mathbf{v} \oplus \epsilon) - \max_{\mathbf{v}_i^- \in \mathbf{V}^-} \overline{p_{\mathbf{v}_i^-, h}}(\mathbf{v} \oplus \epsilon) > 2\Delta, \tag{8}$$

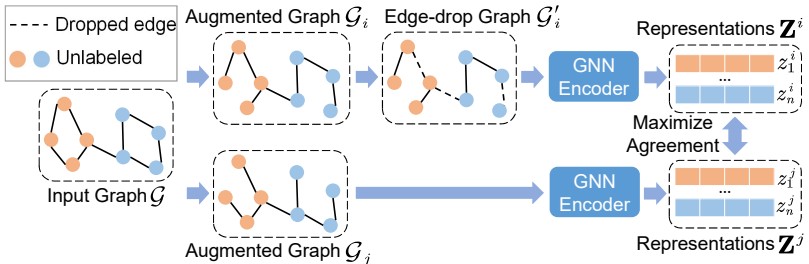

Figure 1: General Framework of training GNN encoder via RES.

where $\Delta = 1 - \frac{\binom{d}{e}}{\binom{d+k}{e}} \cdot \beta^k$ and $e = \|\mathbf{v} \oplus \epsilon\|_0$ denotes the number of remaining edges of $\mathbf{v}$ after injecting $\epsilon$. Then, with a confidence level of at least $1 - \alpha$, we have:

$$p_{\mathbf{v}^+, h}(\mathbf{v}' \oplus \epsilon) > \max_{\mathbf{v}_i^- \in \mathbf{V}^-} p_{\mathbf{v}_i^-, h}(\mathbf{v}' \oplus \epsilon). \tag{9}$$

The proof is in Appendix C.2. Theorem 2 theoretically guarantees that of $\mathbf{v}'$ is still the positive sample of $\mathbf{v}^+$ with $(1 - \alpha)$ confidence if the worst-case margin of $\mathbf{v} \oplus \epsilon$ under $h$ is larger than $2\Delta$, which indicates that the certified robustness of $h$ at $(\mathbf{v}, \mathbf{v}^+)$ is holds in this case. Moreover, it also paves us a efficient way to compute the certified perturbation size in practical (See in Sec. 5.2).

### 4.3 Transfer the Certified Robustness to Downstream Tasks

Though we have theoretically shown the capability of learning $l_0^k$-certifiably robust node/graph representations with RES, it is unclear whether the certified robustness of the smoothed GNN encoder can be preserved in downstream tasks. To address this concern, we propose a theorem that establishes a direct relationship between the certified robustness of GCL and that of downstream tasks.

**Theorem 3.** *Given a GNN encoder $h$ trained via GCL and an clean input $\mathbf{v}$. $\mathbf{v}^+$ and $\mathbf{V}^- = \{\mathbf{v}_1^-, \cdots, \mathbf{v}_n^-\}$ are the positive sample and the set of negative samples of $\mathbf{v}$, respectively. Let $c^+$ and $c_i^-$ denote the latent classes of $\mathbf{v}^+$ and $\mathbf{v}_i^-$, respectively. Suppose $f$ is the downstream classifier that classify a data point into one of the classes in $\mathcal{C}$. Then, we have*

$$\mathbb{P}(f(h(\mathbf{v})) = c^+) > \max_{\mathbf{v}_i^- \in \mathbf{V}^-} \mathbb{P}(f(h(\mathbf{v})) = c_i^-) \tag{10}$$

The proof is in Appendix C.3. By applying Theorem 3, we can prove given $\mathbf{v}$'s perturbed version $\mathbf{v}' = \mathbf{v} \oplus \delta$, where $||\delta||_0 \le k$, if $\mathbf{v}$ and $\mathbf{v}'$ satisfy Eq. (8) and Eq. (9) in Theorem 2, we have

$$\mathbb{P}(f(h(\mathbf{v}' \oplus \epsilon)) = c^+) > \max_{\mathbf{v}_i^- \in \mathbf{V}^-} \mathbb{P}(f(h(\mathbf{v}' \oplus \epsilon)) = c_i^-), \ \ \forall \|\delta\|_0 \le k, \tag{11}$$

which implies provable $l_0^k$-certified robustness retention of $h$ at $(\mathbf{v}, c^+)$ in downstream tasks.

## 5 Practical Algorithms

In this section, we first propose a simple yet effective GCL method to train the GNN encoder $h$ robustly. Then we introduce the practical algorithms for the prediction and robustness certification.

### 5.1 Training Robust Base Encoder

Theorem 2 holds regardless how the base GNN encoder $h$ is trained. However, introducing randomized edgedrop noise solely to test samples during the inference phase could potentially compromise performance in downstream tasks, which further negatively impact the certified robustness performance based on Eq. 8. In order to make $g$ can classify and certify $(\mathbf{v}, c^+)$ correctly and robustly, inspire by [45], we propose to train the base GNN encoder $h$ via GCL with randomized edgedrop noise. Our key idea is to inject randomized edgedrop noise to one augmented view and maximize the probability in Eq. 5 by using GCL to maximize the agreement between two views. Fig. 1 gives an illustration of general process of training the base GNN encoder via our RES. Given an input graph $\mathcal{G}$, two contrasting views $\mathcal{G}_i$ and $\mathcal{G}_j$ are generated from $\mathcal{G}$ through stochastic data augmentations. Specifically, two views generated from the same instance are usually considered as a positive pair, while two views constructed from different instances are considered as a negative pair. After that, we inject the randomized edgedrop noise $\epsilon$ to one of the two views $\mathcal{G}_i$ and obtain $\mathcal{G}_i'$. Finally, two GNN encoders are used to generate embeddings of the views and a contrastive loss is applied to maximize the agreement between $\mathcal{G}_i'$ and $\mathcal{G}_j$. The training algorithm is summarized in Appendix D.

## 5.2 Prediction & Certification

Following [46, 24], we then present the algorithm to use the smoothed GNN encoder $g(\mathbf{v})$ in the downstream tasks and derive the robustness certificates through Monte Carlo algorithms.

**Prediction on Downstream Tasks.** We draw $\mu$ samples of $h(\mathbf{v} \oplus \epsilon)$, corrupted by randomized edgedrop noises. Then we obtain their predictions via the downstream classifier. If $c_A$ is the class which has the largest frequency $\mu_{c_A}$ among the $\mu$ predictions, $c_A$ is returned as the final prediction.

**Compute Robustness Certificates.** One of robustness certificates is the certified perturbation size, which is the maximum attack budget that do not change the prediction of an instance no matter what perturbations within the budget are used. To derive the certified perturbation size of an instance $\mathbf{v}$, we need to estimate the lower bound $\underline{p_{\mathbf{v}^+,h}}(\mathbf{v} \oplus \epsilon)$ and upper bound $\overline{p_{\mathbf{v}_i^-,h}}(\mathbf{v} \oplus \epsilon)$ in Eq. 8. Since it is challenging to directly estimate them in GCL, inspired by Theorem 3, if the prediction of $h(\mathbf{v} \oplus \epsilon)$ in downstream tasks is correct, $\mathbf{v} \oplus \epsilon$ will also be the positive sample of $\mathbf{v}$ under $h$, which indicates the connection between $\underline{p_{\mathbf{v}^+,h}}(\mathbf{v} \oplus \epsilon)$ and the probability of correctly classified in the downstream tasks. Formally, given a label set $\mathcal{C}$ for the downstream task, let $\underline{p_A}$ denotes the lower bound of the probability that $h(\mathbf{v} \oplus \epsilon)$ is correctly classified as $c_A \in \mathcal{C}$ in the downstream task, $\underline{p_B}$ is the upper bound of the probability that $h(\mathbf{v} \oplus \epsilon)$ is classified as $c$ for $c \in \mathcal{C} \backslash \{c_A\}$. we have the following probability bound by a confidence level at least $1 - \alpha$:

$$\underline{p_{\mathbf{v}^+,h}}(\mathbf{v} \oplus \epsilon) = \underline{p_A} = B(\frac{\alpha}{|\mathcal{C}|}; \mu_{c_A}, \mu - \mu_{c_A} + 1), \quad \overline{p_{\mathbf{v}_i^-,h}}(\mathbf{v} \oplus \epsilon) = \overline{p_B} = \min\left(\max_{c \neq c_A} \overline{p_c}, 1 - \underline{p_A}\right) \quad (12)$$

where $\overline{p_c} = B(1 - \frac{\alpha}{|\mathcal{C}|}; \mu_c + 1, \mu - \mu_c), \forall c \neq c_A$, and $B(q; u, w)$ is the $q$-th quantile of a beta distribution with shape parameter $u$ and $w$. After that, we calculate $\Delta$ based on Theorem 2 and the maximum $k$ that satisfies Eq. 8 is the certified perturbation size.

# 6 Experiments

In this section, we conduct experiments to answer the following research questions: (**Q1**) How robust is RES under various adversarial attacks? (**Q2**) Can RES effectively provide certifiable robustness for various GCL methods? (**Q3**) How does RES contribute to the robustness in GCL?

## 6.1 Experimental Setup

**Datasets.** We conduct experiments on 4 public benchmark datasets for node classification, i.e., Cora, Pubmed [47], Coauthor-Physics [48] and OGB-arxiv [49], and 3 widely used dataset for graph classification i.e., MUTAG, PROTEINS [50] and OGB-molhiv [49]. We use public splits for Cora and Pubmed, and for five other datasets, we perform a 10/10/80 random split for training, validation, and testing, respectively. The details and splits of these datasets are summarized in Appendix G.1.

**Attack Methods.** To demonstrate the robustness of RES to various structural noises, we evaluate RES on 4 types of structural attacks in an evasion setting, i.e., Random attack, Nettack [11], PRBCD [51], CLGA [16] for both node and graph classification. In our evaluation, we first train a GNN encoder on a clean dataset using GCL, and then subject RES to attacks during the inference phase by employing perturbed graphs in downstream tasks. Especially, CLGA is an poisoning attack methods for GCL. To align it with our evasion setting, we directly employ the poisoned graph generated by CLGA in downstream tasks for evaluation. The details of these attacks are given in Appendix G.2.

**Compared Methods.** We employ 4 state-of-the-art GCL methods, i.e., GRACE [30], BGRL [52], DGI [31] and GraphCL [9], as baselines. More specifically, GRACE, BGRL and DGI are for node classification, GraphCL and BGRL are for graph classification. We apply our RES on them to train the smoothed GNN encoders. Recall one of our goal is to validate that our method can enhance the robustness of GCL against structural noises, we also consider two robust GCL methods (i.e., Ariel [15] and GCL-Jaccard [12]) and two unsupervised methods ( i.e., Node2Vec [53] and GAE [54]) as the baselines. All hyperparameters of the baselines are tuned based on the validation set to make fair comparisons. The detailed descriptions of the baselines are given in Appendix G.3

**Evaluation Protocol.** In this paper, we conduct experiments on both transductive node classification and inductive graph classification tasks. For each experiment, a 2-layer GCN is employed as the backbone GNN encoder. We adopt the common used linear evaluation scheme [31], where each model is firstly trained via GCL and then the resulting embeddings are used to train and test a

Table 1: Robust accuracy results for node classification.

| Dataset | Graph | Node2Vec | GAE | GCL-Jac. | Ariel | GRACE | RES-GRACE |
|---|---|---|---|---|---|---|---|
| Cora | Raw | 67.6±1.0 | 76.8±0.9 | 76.1±2.0 | **79.8±0.6** | 77.1±1.6 | 79.7±1.0 |
| | Random | 57.7±0.7 | 74.2±0.9 | 74.1±2.0 | 76.3±0.6 | 74.5±2.1 | **79.1±1.2** |
| | CLGA | 64.7±0.7 | 72.7±1.3 | 73.7±1.2 | 76.6±0.4 | 74.9±2.0 | **78.2±1.0** |
| | PRBCD | 63.3±3.2 | 75.6±2.1 | 75.3±1.2 | 75.6±0.2 | 75.8±2.5 | **78.5±1.7** |
| Pubmed | Raw | 66.4±2.0 | 78.4±0.4 | 78.2±2.7 | 78.0±1.1 | 79.5±2.9 | **79.5±1.2** |
| | Random | 56.8±1.4 | 71.8±1.0 | 75.8±2.4 | 75.9±0.2 | 75.0±1.0 | **78.2±0.9** |
| | CLGA | 61.9±1.2 | 77.6±0.5 | 76.2±2.8 | 77.5±1.1 | 76.6±2.5 | **78.3±1.1** |
| | PRBCD | 55.9±1.5 | 74.8±2.5 | 73.1±2.0 | 73.5±1.6 | 73.2±2.3 | **78.8±1.7** |
| Physics | Raw | 92.9±0.1 | 95.2±0.1 | 94.1±0.3 | **95.3±0.3** | 94.0±0.4 | 94.7±0.2 |
| | Random | 85.7±0.3 | 93.7±0.1 | 93.4±0.3 | 93.8±0.1 | 92.6±0.5 | **94.2±0.3** |
| | PRBCD | 81.8±0.6 | 91.1±0.7 | 91.6±0.2 | 91.0±0.9 | 89.2±0.6 | **94.1±0.2** |
| OGB-arxiv | Raw | 64.6±0.1 | 61.5±0.5 | 64.7±0.2 | 64.7±0.3 | 65.1±0.5 | **65.2±0.1** |
| | Random | 52.4±0.1 | 57.4±0.5 | 59.0±0.1 | 59.4±0.4 | 59.0±0.2 | **60.0±0.1** |
| | PRBCD | 56.5±0.5 | 54.1±0.5 | 56.5±0.4 | 57.0±0.9 | 55.7±0.4 | **58.3±0.4** |

(a) Cora     (b) OGB-arxiv     (c) MUTAG     (d) PROTEINS

Figure 2: Certified accuracy of smoothed GCL

$l_2$-regularized logistic regression classifier. The certified accuracy and robust accuracy on test nodes are used to evaluate the robustness performance. Specifically, *certified accuracy* [46, 24] denotes the fraction of correctly predicted test nodes/graphs whose certified perturbation size is no smaller than the given perturbation size. Each experiment is conduct 5 times and the average results are reported.

## 6.2 Performance of Robustness

To answer **Q1**, we compare RES with the baselines on various noisy graphs. We also conduct experiments to demonstrate that RES can improve the robustness against different noisy levels, which can be found in Appendix I.2. We focus on node classification as the downstream task on four types of noisy graphs, i.e., raw graphs, random attack perturbed graphs, CLGA perturbed graphs and PRBCD perturbed graphs. The perturbation rate of noisy graphs is 0.1. The details of the noisy graphs are presented in Appendix G.2. GRACE is used as the target GCL method to train a GCN encoder. The smoothed version of GRACE is denoted as RES-GRACE. The results on Cora, Coauthor-Physics and OGB-arxiv are given in Table 1. From the table, we observe: **(i)** When no attack is applied on the clean graph, our RES-GRACE achieve state-of-the-art performance, especially on large-scale datasets, which indicates RES is beneficial to learn good representation by injecting random edgedrop noise to the graph. **(ii)** The structural noises degrade the performances of all baselines. However, its impact to RES-GRACE is negligible. RES-GRACE outperforms all baselines including two robust GCL methods, which indicates RES could eliminate the effects of the noisy edges. More results on graph classification are shown in Appendix I.1.

## 6.3 Performance of Certificates

To answer **Q2**, we use certified accuracy as the metric to evaluate the performance of robustness certificates. We choose GCN as the GNN encoder and employ our method on GRACE [30] and GraphCL [9] for node classification and graph classification, respectively. We select the overall test nodes as the target nodes and set $\mu = 200$, $(1 - \alpha) = 99.9\%$ to compute the certified accuracy. The results for various $\beta$ are shown in Fig. 2, where the x-axis denotes the given perturbation size. From the figures, we can observe that $\beta$ controls the tradeoff between robustness and model utility. When $\beta$ is larger, the certified accuracy on clean graph is larger, but it drops more quickly as the perturbation size increases. Especially, when $\beta = 0.999$, the certified accuracy is nearly independent of the perturbation size. Our analysis reveals that when $\beta$ is low, more structural information is retained on the graph, which benefits the performance of RES under no attack, but also results in more noisy edges being retained on the noisy graph, leading to lower certified accuracy. Conversely, when $\beta$ is large, less structural information is retained on the graph, which may affect the performance of RES

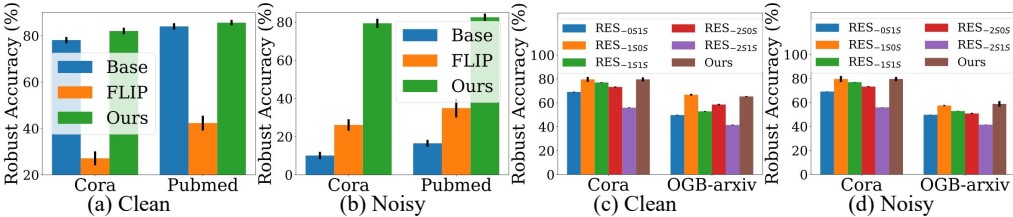

Figure 3: Ablation Study Results: Comparisons (a) and (b) with FLIP, (c) and (d) with $\text{RES}_{i\mathbf{S}j\mathbf{S}}$ under no attack, but also results in fewer noisy edges remaining on the noisy graph, leading to higher certified accuracy for larger perturbation sizes.

## 6.4 Ablation Study

To answer **Q3**, we conduct several ablation studies to investigate the effects of our proposed RES method. More results of ablation studies are shown in Appendix J. We also investigate how hyperparameters $(1 - \alpha)$ and $\mu$ affect the performance of our RES, which can be found in Appendix K.

Our first goal is to demonstrate RES is more effective in GCL compared with vanilla binary randomized smoothing in [24], We implement a variant of our model, FLIP, which replaces the random edgedrop noise with binary random noise [24], thereby flipping the connection status within the graph with the probability $\beta$. For our method, we set $\beta = 0.9$, $1 - \alpha = 99\%$, and $\mu = 50$. For FLIP, to ensure a fair comparison, we set $\alpha = 99\%$ and $\mu = 50$, and vary $\beta$ over $\{0.1, 0.2, \cdots, 0.9\}$ and select the value which yields the best performance on the validation set of clean graphs. The target GCL method is selected as GRACE. We compare the robust accuracy of the two methods on the clean graph and noisy graph under Nettack with attack budget 3. The average robust accuracy and standard deviation on Cora and Pubmed are reported in Fig. 3 (a) and (b). We observe: **(i)** RES achieves better results on various graphs (i.e., clean and noisy graph) compared to FLIP and the base GCL method. **(ii)** FLIP performs much worse than the base GCL method and RES on the clean graph, suggesting that vanilla binary randomized smoothing is ineffective in GCL. That is because FLIP introduces excessive noisy edges, which in turn makes it challenging for GCL to learn accurate representations.

Second goal is to understand how RES contributes to the robustness of GCL. We implement several variants of our model by removing structural information in the training and testing phases, which are named $\text{RES}_{i\mathbf{S}j\mathbf{S}}$, where $i \in \{0, 1, 2\}$ and $j \in \{0, 1\}$ denote the number of removed structures in the training and testing phases, respectively. For our method, we set $\beta = 0.9$, $1 - \alpha = 99\%$ and $\mu = 50$. We compare the robust accuracy on clean and noisy graphs and use PRBCD to perturb 10% of the total number of edges in the graph (before attack) for noisy graphs. The overall test set was selected as the target nodes. The results on Cora and OGB-arxiv are shown in Fig. 3 (c) and (d). We observe: **(i)** Our method significantly outperforms the ablative methods on various graphs, corroborating the effectiveness of randomized edgedrop smoothing for GCL. **(ii)** Our method and $\text{RES}_{1\mathbf{S}0\mathbf{S}}$ achieve better robust accuracy on various graphs compared to other ablative methods. This is because dropping edges in only one augmented view during training both alleviates over-fitting and increases the worst-case margin, which is helpful for robustness and model utility. **(iii)** $\text{RES}_{2\mathbf{S}j\mathbf{S}}$ and $\text{RES}_{i\mathbf{S}1\mathbf{S}}$ perform worse than other methods because of the absence of structural information in training and test phases, highlighting the importance of structural information in GCL. Moreover, the ablation studies on the effectiveness of our approach for training the GNN encoder are in Appendix J.1.

## 7 Conclusion

In this paper, we present the first work to study the certifiable robustness of GCL. We address the existing ambiguity in quantifying the robustness of GCL to perturbations by introducing a unified definition for robustness in GCL. Our proposed approach, Randomized Edgedrop Smoothing, injects randomized edgedrop noise into graphs to provide certified robustness for GCL on unlabeled data, while minimizing the introduction of spurious edges. Theoretical analysis establishes the provable robust performance of our encoder in downstream tasks. Additionally, we present an effective training method for robust GCL. Extensive empirical evaluations on various real-world datasets show that our method guarantees certifiable robustness and enhances the robustness of any GCL model.

# 8 Acknowledgements

This material is based upon work supported by, or in part by, the National Science Foundation (NSF) under grant number IIS-1909702, the Army Research Office (ARO) under grant number W911NF21-1-0198, and Department of Homeland Security (DNS) CINA under grant number E205949D. The findings in this paper do not necessarily reflect the view of the funding agencies.

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

# A Details of Related Works

## A.1 Graph Contrastive Learning

GCL has recently gained significant attention and shows promise for improving graph representations in scenarios where labeled data is scarce [31, 32, 9, 30, 27, 15, 33, 32]. Generally, GCL creates two views through data augmentation and contrasts representations between two views. GraphCL [9] focuses on graph classification by exploring four types of augmentations including node dropping, edge perturbation, attribute masking and subgraph sampling. GRACE [30] and GCA [27] adapt SimCLR [55] to graphs to maximize the mutual information between two views for each node through a variety of data augmentation. DGI [31] applied InfoMax principle [56] to train a GNN encoder by maximizing the mutual information between node-level and graph-level representations. MVGRL [32] proposes to learn representations by maximizing the mutual information between the cross-view representations of nodes and graphs. Several recent works [16, 36] show that GCL is vulnerable to adversarial attacks. CLGA [16] is an unsupervised poisoning attacks for attack graph contrastive learning. In detail, the gradients of the adjacency matrices for both views are computed, and edge flipping is performed using gradient ascent to maximize the contrastive loss. Despite very few empirical works [9, 15, 14] on robustness of GCL, there are no existing works studying the certified robustness of GCL. In contrast, we propose to provide robustness certificates for GCL by using randomized edgedrop smoothing. To the best of our knowledge, our method is the first work to study the certified robustness of GCL.

## A.2 Certifiable Robustness of Graph Neural Networks

Several recent studies investigate the certified robustness of GNNs in the supervised setting [23, 28, 29, 26, 24, 25, 37]. Zügner et al. [23] are the first to explore certifiable robustness with respect to node feature perturbations. Subsequent works[28, 26, 29, 24, 38] extend the analysis to certifiable robustness under topological attacks. For instance, Bojchevski et al. [28] propose a branch-and-bound algorithm to achieve tight bounds on the global optimum of certificates for topological attacks. Bojchevski et al. [29] further adapt the randomized smoothing technique to sparse settings, deriving certified robustness for GNNs. This approach involves injecting random noise into test samples to mitigate the negative effects of adversarial perturbations. Wang et al. [24] further refine this technique to provide theoretically tight robust certificates. Our work is inherently different from them: (**i**) existing work focuses on the certified robustness of GNN under (semi)-supervised setting; while we provide a unified definition to evaluate and certify the robustness of GCL for unsupervised representation learning. (**ii**) we theoretically provide the certified robustness for GCL in the absence of labeled data, and this certified robustness can provably sustained in downstream tasks. (**iii**) we design an effective training method to enhance the robustness of GCL.

# B Preliminary of Randomized Smoothing

One potential solution for achieving certified robustness in GNNs is binary randomized smoothing presented in [24]. Specifically, consider a noisy vector $\epsilon$ in the discrete space $\{0, 1\}^N$

$$\mathbb{P}(\epsilon_i = 0) = \beta, \quad \mathbb{P}(\epsilon_i = 1) = 1 - \beta, \tag{B.1}$$

where $i = 1, 2, \cdots, N$. This indicates that the connection status of the $i$-th entry of $\mathbf{v}$ will be flipped with probability $1 - \beta$ and preserved with probability $\beta$. Given a base node or graph classifier $f(\mathbf{v})$ that returns the class label with highest label probability, the smoothed classifier $g$ is defined as:

$$g(\mathbf{v}) = \arg\max_{y \in \mathcal{Y}} \mathbb{P}(f(\mathbf{v} \oplus \epsilon) = y), \tag{B.2}$$

where $\mathcal{Y}$ denotes the label set, and $\oplus$ represents the XOR operation between two binary variables, which combines the structural information of both variables. $\mathbb{P}(f(\mathbf{v} \oplus \epsilon) = y)$ is the probability that the base classifier $f$ predicts class $y$ when random noise $\epsilon$ is added to $\mathbf{v}$. According to [24], by injecting multiple $\epsilon$ to $\mathbf{v}$ and returning the most likely class $y_A$ (the majority vote), if it is holds that:

$$\min \mathbb{P}(f(\mathbf{v} \oplus \epsilon) = y_A) \geq \max_{y \neq y_A} \mathbb{P}(f(\mathbf{v} \oplus \epsilon) = y), \text{ s.t. } \|\epsilon\|_0 \leq k, \tag{B.3}$$

where $y_A$ is the class with the highest label probability. Then we can guarantee there exists a certified perturbation size $k$ such that for any perturbation $\delta$ with $\|\delta\|_0 \leq k$, the predictions of $\mathbf{v} \oplus \delta$ will

remain unchanged, that is, $g(\mathbf{v} \oplus \delta) = f(\mathbf{v} \oplus \delta \oplus \epsilon) = f(\mathbf{v} \oplus \epsilon) = y_A$. Thus, $f$ is certifiably robust at $(\mathbf{v}, y_A)$ when perturbing at most $k$ edges. For more deatils of the proof, please refer to [24].

However, directly applying the vanilla randomized smoothing approach to certify robustness in GCL is inapplicable. This approach may require injecting random noise into the entire graph during data augmentation, resulting in an excessive number of noisy/spurious edges that can significantly harm downstream task performance, particularly for large and sparse graphs. In contrast, our proposed method, Randomized Edgedrop Smoothing (RES), aims to prevent the introduction of excessive spurious edges in graphs. RES achieves this by injecting randomized edgedrop noises, where observed edges are randomly removed from the graphs with a certain probability. Additionally, we propose an effective training method for robust GCL. Our experimental results in Sec. 6.4 demonstrate that our method significantly outperforms the ablative method, FLIP, that directly applies vanilla randomized smoothing [24] in GCL. Specifically, on the clean Cora and Pubmed graphs, our method achieves accuracies of 82% and 85.6% respectively. On the Nettack perturbed [11] Cora and Pubmed graphs, our method achieves robust accuracies of 79.4% and 82.5%, respectively. In contrast, FLIP achieves only up to 27% and 42% robust accuracies on the Cora and Pubmed datasets, respectively. These results strongly validate the effectiveness of our RES approach,

## C    Detailed Proofs

### C.1    Proof of Theorem 1

To begin, we provide a formalization of *similarity* in relation to the latent class.

Given a GNN encoder $h$ and an input sample $\mathbf{v}$, which consists of a positive sample $\mathbf{v}^+$ and a negative sample $\mathbf{v}^-$ an input sample $\mathbf{v}$ with positive sample $\mathbf{v}^+$ and negative sample $\mathbf{v}^-$, according to Def. 1, we assume that $c^+, c^- \in \mathcal{C}$ represent the latent classes of $\mathbf{v}^+$ and $\mathbf{v}^-$ in the latent space of $h$, respectively. These latent classes are based on the distribution $\eta$ over $\mathcal{C}$. we assume that $c^+, c^- \in \mathcal{C}$ are the latent class of $\mathbf{v}^+$ and $\mathbf{v}^-$ in the latent space of $h$, respectively, which are randomly determined based on the distribution $\eta$ on $\mathcal{C}$. Similar data $h(\mathbf{v}), h(\mathbf{v}^+)$ are i.i.d. draws from the same class distribution $\mathcal{D}_{c^+}$, whereas negative samples originate from the marginal of $\mathcal{D}_{sim}$, which are formalized as follow:

$$\mathcal{D}_{sim}(\mathbf{v}, \mathbf{v}^+) = \mathbb{E}_{c^+ \sim \eta} \mathcal{D}_{c^+}(\mathbf{v}) \mathcal{D}_{c^+}(\mathbf{v}^+).$$
$$\mathcal{D}_{neg}(\mathbf{v}^-) = \mathbb{E}_{c^- \sim \eta} \mathcal{D}_{c^-}(\mathbf{v}^-). \tag{C.4}$$

Since classes are allowed to overlap and/or be fine-grained [39], this is a plausible formalization of "similarity". The formalization connects similarity with latent class, thereby offering a viable way to employ the similarity between samples under the latent space $h$ for defining the probability of $\mathbf{v}$ being the positive sample of $\mathbf{v}^+$.

**Lemma 1** (The Generalized Extreme Value Distribution [44]). Let $\mathbf{v}_1, \mathbf{v}_2, \cdots$ be a sequence of i.i.d samples from a common distribution function $F$. Let $M_N = \max\{\mathbf{v}_1, \cdots, \mathbf{v}_N\}$, indicting the maximum of $\{\mathbf{v}_1, \cdots, \mathbf{v}_N\}$. Then if $\{a_N > 0\}$ and $\{b_N \in \mathbb{R}\}$ are sequences of constants such that

$$\mathbb{P}\{(M_N - b_N)/a_N \le z\} \to G(z), \tag{C.5}$$

where $G$ is a non-degenerate distribution function. It is a member of the generalized extreme value family of distributions, which belongs to either the Gumbel family (Type I), the Fréchet family (Type II) or the Reverse Weibull family (Type III) with their CDFs as follows:

$$\text{Gumbel family (Type I):} \quad G(z) = \exp\{-\exp[-(\frac{z-b}{a})]\}, \quad z \in \mathbb{R},$$

$$\text{Fréchet family (Type II):} \quad G(z) = \begin{cases} 0, & z<b, \\ \exp\{-(\frac{z-b}{a})^{-\sigma}\}, & z \ge b, \end{cases} \tag{C.6}$$

$$\text{Reverse Weibull family (Type III):} \quad G(z) = \begin{cases} \exp\{-(\frac{b-z}{a})^{-\sigma}\}, & z<b, \\ 1, & z \ge b, \end{cases}$$

where $a > 0$, $b \in R$ and $\sigma > 0$ are the scale, location and shape parameters, respectively.

**Theorem 1.** *Let $\mathbb{B}(\mathbf{v}^+)$ be a space around $\mathbf{v}^+$ as defined in Eq. 4. Given an input sample $\mathbf{v}$ and a GNN encoder $h$ learnt by GCL in Eq. 1, the probability of $\mathbf{v}$ being the positive sample of $\mathbf{v}^+$ is:*

$$Pr(\mathbf{v} \in \mathbb{B}(\mathbf{v}^+); h) = \exp\left[-\left(\frac{1 - s(h(\mathbf{v}), h(\mathbf{v}^+))}{a}\right)^\sigma\right], \tag{C.7}$$

*where $s(\cdot, \cdot)$ is the cosine similarity function. $a, \sigma > 0$ are Weibull shape and scale parameters [44].*

*Proof.* We first assume $\{h(\mathbf{v}_1), h(\mathbf{v}_2), \cdots\}$ are i.i.d samples under the latent space $h$. Then, suppose $\mathbf{V}^- = \{\mathbf{v}_1^-, \cdot, \mathbf{v}_n^-\}$ is the set of negative samples of $\mathbf{v}$. We assume there exists a continuous non-degenerate margin distribution $M$, which is denoted as follows:

$$M := \min_{i \in [n]} D_i, \quad \text{with } D_i := (1 - s(h(\mathbf{v}^+), h(\mathbf{v}_i^-)))/2, \tag{C.8}$$

where $M, D_i \in [0, 1]$, and $s(\cdot, \cdot)$ is the cosine similarity function. $M$ indicates the half of the minimum distance between $h(\mathbf{v}^+)$ and $h(\mathbf{v}_i^-)$.

According to Lemma 1, we know that there exists $G(z)$ in the three types of generalized extreme value family. And since Lemma 1 is applied to the maximum, refer to Eq. (C.8), we transfer the variable $M$ to $\overline{M} := \max_{i \in [n]} -D_i$, Since $-D_i$ is bounded $(-D_i < 0)$, let $b = 0$, the marginal distribution of $D_i$ series, for $i = 1, 2, \cdots$, can be Reverse Weibull family. Therefore, the asymptotic marginal distribution of $\overline{M}$ fits into the Reverse Weibull distribution:

$$G(z) = \begin{cases} \exp\{-(\frac{-z}{a})^{-\sigma}\}, & z<0, \\ 1, & z\geq 0, \end{cases}$$

where $\sigma > 0$ is the shape parameter and $a$ is the scale parameter. Compared to Eq. (C.6), here, $b = 0$ because $-D_i$ is bound $(-D_i < 0)$. We use margin distances $D_i$ of the $\lambda$ closest samples with $\mathbf{v}^+$ to estimate the parameters $\alpha$ and $\sigma$, which means to estimate $\widehat{W}$ of the distribution function $W$.

Following Eq. (C.4), if the similarity $s(h(\mathbf{v}), h(\mathbf{v}^+))$ is larger, $\mathbf{v}$ is more possible to be from the same latent class as $\mathbf{v}^+$, which implies $\mathbf{v}$ is the positive sample of $\mathbf{v}^+$. Since the distance between $h(\mathbf{v})$ and $h(\mathbf{v}^+)$ can be denoted as $1 - s(h(\mathbf{v}^+), h(\mathbf{v}_i^-))$, the probability of $\mathbf{v}$ is the positive sample of $\mathbf{v}^+$ can be written as:

$$\begin{aligned}
&\mathbb{P}(\mathbf{v} \in \mathbb{B}(\mathbf{v}^+); h) \\
=&\mathbb{P}(1 - s(h(\mathbf{v}), h(\mathbf{v}^+)) < \min\{D_1, \cdots, D_n\}) \\
=&\mathbb{P}(s(h(\mathbf{v}), h(\mathbf{v}^+) - 1) > \max\{-D_1, \cdots, -D_n\}) \\
=&\mathbb{P}(\overline{M} < s(h(\mathbf{v}), h(\mathbf{v}^+)) - 1) \\
=&\widehat{W}(s(h(\mathbf{v}), h(\mathbf{v}^+)) - 1).
\end{aligned} \tag{C.9}$$

Let $b_n = 0$, $a_n = 0$ and $z = s(h(\mathbf{v}), h(\mathbf{v}^+)) - 1$, since $z = s(h(\mathbf{v}), h(\mathbf{v}^+)) - 1 < 0$, we can rewrite Eq. (C.9) as (C.7) according to Lemma 1. $\square$

## C.2 Proof of Theorem 2

For simplicity of notation, let $p_{\mathbf{v}^+, h}(\mathbf{v}) = \mathbb{P}(\mathbf{v} \in \mathbb{B}(\mathbf{v}^+); h)$. Then, given a randomized edgedrop noise $\epsilon \in \mathcal{D}_\epsilon$ with the following probability distribution:

$$\mathbb{P}(\epsilon_i = 0 | \mathbf{v}_i = 0) = 1, \ \mathbb{P}(\epsilon_i = 0 | \mathbf{v}_i = 1) = \beta, \text{ and } \mathbb{P}(\epsilon_i = 1 | \mathbf{v}_i = 1) = 1 - \beta, \tag{C.10}$$

where $\epsilon_i$ is the $i$-entry of noisy vector $\epsilon$ and $\mathbf{v}_i$ is the connection status of $i$-th entry of $\mathbf{v}$. Then, we let

$$\underline{p_{\mathbf{v}^+, h}}(\mathbf{v} \oplus \epsilon) = \inf_{\epsilon \in \mathcal{D}_\epsilon} p_{\mathbf{v}^+, h}(\mathbf{v} \oplus \epsilon), \ \overline{p_{\mathbf{v}^+, h}}(\mathbf{v} \oplus \epsilon) = \sup_{\epsilon \in \mathcal{D}_\epsilon} p_{\mathbf{v}^+, h}(\mathbf{v} \oplus \epsilon), \tag{C.11}$$

where $\underline{p_{\mathbf{v}^+, h}}(\mathbf{v} \oplus \epsilon)$ and $\overline{p_{\mathbf{v}_i^-, h}}(\mathbf{v} \oplus \epsilon)$ denote the lower bound and the upper bound on $p_{\mathbf{v}^+, h}(\mathbf{v} \oplus \epsilon)$, respectively. The formal theorem is presented as follows:

**Theorem 2.** *Let $\mathbf{v}$ be a clean input and $\mathbf{v}' = \mathbf{v} \oplus \delta$ be its perturbed version, where $||\delta||_0 \leq k$. $\mathbf{V}^- = \{\mathbf{v}_1^-, \cdots, \mathbf{v}_n^-\}$ is the set of negative samples of $\mathbf{v}$. If for all $\mathbf{v}_i^- \in \mathbf{V}^-$:*

$$\underline{p_{\mathbf{v}^+, h}}(\mathbf{v} \oplus \epsilon) - \max_{\mathbf{v}_i^- \in \mathbf{V}^-} \overline{p_{\mathbf{v}_i^-, h}}(\mathbf{v} \oplus \epsilon) > 2\Delta, \tag{C.12}$$

*where*

$$\Delta = 1 - \frac{\binom{d}{e}}{\binom{d+k}{e}} \cdot \beta^k, \tag{C.13}$$

*and $e = \|\mathbf{v} \oplus \epsilon\|_0$ denotes the number of remaining edges of $\mathbf{v}$ after injecting $\epsilon$, then with a confidence level of at least $1 - \alpha$, we have:*

$$p_{\mathbf{v}^+,h}(\mathbf{v}' \oplus \epsilon) > \max_{\mathbf{v}_i^- \in \mathbf{V}^-} p_{\mathbf{v}_i^-,h}(\mathbf{v}' \oplus \epsilon). \tag{C.14}$$

*Proof.* Suppose $\mathbf{V} = \{\mathbf{v}^+, \mathbf{v}_1^-, \cdots, \mathbf{v}_n^-\}$ is the set of positive and negative samples of $\mathbf{v}$. For any $\mathbf{v}_i \in \mathbf{V}$, let $p_{\mathbf{v}_i,h}(\mathbf{v}) = \mathbb{P}(\mathbf{v} \in \mathbb{B}(\mathbf{v}_i); h)$ for simplicity, we have:

$$\begin{aligned} p_{\mathbf{v}_i,h}(\mathbf{v} \oplus \epsilon) &= \mathbb{P}(\mathbf{v} \oplus \epsilon \in \mathbb{B}(\mathbf{v}_i); h) \\ p_{\mathbf{v}_i,h}(\mathbf{v}' \oplus \epsilon) &= \mathbb{P}(\mathbf{v}' \oplus \epsilon \in \mathbb{B}(\mathbf{v}_i); h). \end{aligned} \tag{C.15}$$

As stated by the law of total probability, we have:

$$\begin{aligned} p_{\mathbf{v}_i,h}(\mathbf{v} \oplus \epsilon) &= \mathbb{P}([\mathbf{v} \oplus \epsilon \in \mathbb{B}(\mathbf{v}_i); h] \wedge [(\mathbf{v}' \oplus \epsilon) \cap \delta = \emptyset]) \\ &\quad + \mathbb{P}([\mathbf{v} \oplus \epsilon \in \mathbb{B}(\mathbf{v}_i); h] \wedge [(\mathbf{v}' \oplus \epsilon) \cap \delta \neq \emptyset]) \\ p_{\mathbf{v}_i,h}(\mathbf{v}' \oplus \epsilon) &= \mathbb{P}([\mathbf{v}' \oplus \epsilon \in \mathbb{B}(\mathbf{v}_i); h] \wedge [(\mathbf{v}' \oplus \epsilon) \cap \delta = \emptyset]) \\ &\quad + \mathbb{P}([\mathbf{v}' \oplus \epsilon \in \mathbb{B}(\mathbf{v}_i); h] \wedge [(\mathbf{v}' \oplus \epsilon) \cap \delta \neq \emptyset]), \end{aligned} \tag{C.16}$$

where $(\mathbf{v}' \oplus \epsilon) \cap \delta$ represent the intersection of the edge sets $(\mathbf{v}' \oplus \epsilon)$ and $\delta$, that is, the set of edges shared in the two structure vectors. Therefore, for $\mathbf{v}' \oplus \epsilon \cap \delta = \emptyset$, it means that no noisy edge of $\delta$ exists in $(\mathbf{v}' \oplus \epsilon)$, which indicates that $\mathbf{v} \oplus \epsilon$ and $\mathbf{v}' \oplus \epsilon$ are structural identical at all indices, i.e., $\mathbf{v}' \oplus \epsilon = \mathbf{v} \oplus \epsilon$. Therefore, we can then derive the following equality:

$$\mathbb{P}([\mathbf{v} \oplus \epsilon \in \mathbb{B}(\mathbf{v}_i); h] \mid [(\mathbf{v}' \oplus \epsilon) \cap \delta = \emptyset]) = \mathbb{P}([\mathbf{v}' \oplus \epsilon \in \mathbb{B}(\mathbf{v}_i); h] \mid [(\mathbf{v}' \oplus \epsilon) \cap \delta = \emptyset]). \tag{C.17}$$

By multiplying $\mathbb{P}((\mathbf{v}' \oplus \epsilon) \cap \delta = \emptyset)$ in both sides of Eq. (C.17), we have:

$$\mathbb{P}([\mathbf{v} \oplus \epsilon \in \mathbb{B}(\mathbf{v}_i); h] \wedge [(\mathbf{v}' \oplus \epsilon) \cap \delta = \emptyset]) = \mathbb{P}([\mathbf{v}' \oplus \epsilon \in \mathbb{B}(\mathbf{v}_i); h] \wedge [(\mathbf{v}' \oplus \epsilon) \cap \delta = \emptyset]). \tag{C.18}$$

Combining Eq. (C.16) with Eq. (C.18) results in:

$$\begin{aligned} p_{\mathbf{v}_i,h}(\mathbf{v}' \oplus \epsilon) - p_{\mathbf{v}_i,h}(\mathbf{v} \oplus \epsilon) =& \mathbb{P}([\mathbf{v}' \oplus \epsilon \in \mathbb{B}(\mathbf{v}_i); h] \wedge [(\mathbf{v}' \oplus \epsilon) \cap \delta \neq \emptyset]) - \\ & \mathbb{P}([\mathbf{v} \oplus \epsilon \in \mathbb{B}(\mathbf{v}_i); h] \wedge [(\mathbf{v}' \oplus \epsilon) \cap \delta \neq \emptyset]). \end{aligned} \tag{C.19}$$

Since probabilities are non-negative, we can rewrite Eq. (C.19) into the following inequality:

$$\begin{aligned} &p_{\mathbf{v}_i,h}(\mathbf{v} \oplus \epsilon) - \mathbb{P}([\mathbf{v} \oplus \epsilon \in \mathbb{B}(\mathbf{v}_i); h] \wedge [(\mathbf{v}' \oplus \epsilon) \cap \delta \neq \emptyset]) \\ &\leq p_{\mathbf{v}_i,h}(\mathbf{v}' \oplus \epsilon) \leq \\ &p_{\mathbf{v}_i,h}(\mathbf{v} \oplus \epsilon) + \mathbb{P}([\mathbf{v}' \oplus \epsilon \in \mathbb{B}(\mathbf{v}_i); h] \wedge [(\mathbf{v}' \oplus \epsilon) \cap \delta \neq \emptyset]). \end{aligned} \tag{C.20}$$

Applying the conjunction rule, we have:

$$\begin{aligned} &p_{\mathbf{v}_i,h}(\mathbf{v} \oplus \epsilon) - \mathbb{P}((\mathbf{v}' \oplus \epsilon) \cap \delta \neq \emptyset) \leq p_{\mathbf{v}_i,h}(\mathbf{v}' \oplus \epsilon) \leq \\ &p_{\mathbf{v}_i,h}(\mathbf{v} \oplus \epsilon) + \mathbb{P}((\mathbf{v}' \oplus \epsilon) \cap \delta \neq \emptyset). \end{aligned} \tag{C.21}$$

Since $\mathbb{P}((\mathbf{v}' \oplus \epsilon) \cap \delta \neq \emptyset) = 1 - \mathbb{P}((\mathbf{v}' \oplus \epsilon) \cap \delta = \emptyset)$, and $(\mathbf{v}' \oplus \epsilon) \cap \delta = \emptyset$ implies that remaining edges of $\mathbf{v} \oplus \epsilon$ and $\mathbf{v}' \oplus \epsilon$ are identical after injecting the random masking noise, we can derive the following probability:

$$\mathbb{P}((\mathbf{v}' \oplus \epsilon) \cap \delta = \emptyset) = \frac{\binom{d}{e}}{\binom{d+|\delta|}{e}} \cdot \beta^{|\delta|}, \tag{C.22}$$

where $e$ denotes the number of remaining edges, and $k = |\delta|$ denotes the certified perturbation size. $\beta^e (1 - \beta)^{d-e+|\delta|}$ represents the probability that $|\delta|$ noise edges are all dropped and $e$ edges of $\mathbf{v}$ are retained in $\mathbf{v}' \oplus \epsilon$. Therefore, we have:

$$\mathbb{P}((\mathbf{v}' \oplus \epsilon) \cap \delta \neq \emptyset) = 1 - \frac{\binom{d}{e}}{\binom{d+|\delta|}{e}} \cdot \beta^{|\delta|} \leq 1 - \frac{\binom{d}{e}}{\binom{d+k}{e}} \cdot \beta^k = \Delta. \tag{C.23}$$

Substituting Eq. (C.23) into Eq. (C.21), then Eq. (C.21) can be rewritten as

$$p_{\mathbf{v}_i,h}(\mathbf{v} \oplus \epsilon) - \Delta \leq p_{\mathbf{v}_i,h}(\mathbf{v}' \oplus \epsilon) \leq p_{\mathbf{v}_i,h}(\mathbf{v} \oplus \epsilon) + \Delta. \tag{C.24}$$

Referring to Eq. (C.24), for any $\mathbf{v}^+$ and corresponding $\mathbf{v}_i^- \in \mathbf{V}^-$ we have

$$\begin{aligned} p_{\mathbf{v}^+,h}(\mathbf{v}' \oplus \epsilon) &\geq p_{\mathbf{v}^+,h}(\mathbf{v} \oplus \epsilon) - \Delta, \\ p_{\mathbf{v}_i^-,h}(\mathbf{v} \oplus \epsilon) + \Delta &\geq p_{\mathbf{v}_i^-,h}(\mathbf{v}' \oplus \epsilon). \end{aligned} \tag{C.25}$$

Thus, we can derive the following inequality based on Eq. (C.25):

$$\begin{aligned} p_{\mathbf{v}^+,h}(\mathbf{v}' \oplus \epsilon) &\geq p_{\mathbf{v}^+,h}(\mathbf{v} \oplus \epsilon) - \Delta \geq \underline{p_{\mathbf{v}^+,h}}(\mathbf{v} \oplus \epsilon) - \Delta \\ &\geq \max_{\mathbf{v}_i^- \in \mathbf{V}} \overline{p_{\mathbf{v}_i^-,h}}(\mathbf{v} \oplus \epsilon) + \Delta \geq \overline{p_{\mathbf{v}_i^-,h}}(\mathbf{v} \oplus \epsilon) + \Delta \geq p_{\mathbf{v}_i^-,h}(\mathbf{v} \oplus \epsilon) + \Delta \\ &\geq p_{\mathbf{v}_i^-,h}(\mathbf{v}' \oplus \epsilon), \end{aligned} \tag{C.26}$$

which can be restated as Eq. (C.14). This completes our proof. $\qquad\square$

## C.3 Proof of Theorem 3

In order to transfer the certified robustness of GCL to downstream tasks, we first introduce two loss functions, namely, unsupervised loss and supervised loss. Subsequently, we introduce a lemma to establish the relationship between GCL and downstream tasks. Finally, a theorem is proposed to prove that the certified robustness of GCL is provably preserved in downstream tasks.

**Unsupervised Loss**   Given an input sample $\mathbf{v}$ with its positive sample $\mathbf{v}^+$ and $n$ negative samples $\{\mathbf{v}_1^-, \dots, \mathbf{v}_n^-\}$. Let $h : \{0,1\}^n \to \mathbb{R}^d$ be the GNN encoder based on GCL to obtain representations. The unsupervised loss for $h$ at point $\mathbf{v}$ is defined as:

$$L_{un}(\mathbf{v}; h) := \sum_{i=1}^{n} \ell(h(\mathbf{v})^\top (h(\mathbf{v}^+) - h(\mathbf{v}_i^-))), \tag{C.27}$$

where $l$ is logistic loss $l(\mathbf{v}) = \log\left(1 + \sum_{i=1}^{n} \exp\left(-\mathbf{v}_i\right)\right)$ according to [57, 39]. Note that this loss is essentially equivalent to the InfoNCE loss [42, 57] shown in Eq. 1, which is widely-used for GCL.

**Supervised Loss**   Linear evaluation, which learns a downstream linear layer after the base encoder, is a common way to evaluate the performance of GCL model in downstream tasks. Let $\mathcal{C}$ be denoted as the set of latent classes, where $|\mathcal{C}| = m$. We consider the standard supervised learning tasks that classify a data point into one of the classes in $\mathcal{C}$. To connect the GCL task with the downstream classification task, the supervised loss of downstream classifier $f$ at $(\mathbf{x}, y)$ is defined as:

$$L_{sup}(\mathbf{x}, y; f) := \ell(\{f(x)_y - f(x)_{y'}\}_{y' \neq y}), \tag{C.28}$$

where $l$ is the same as the loss function used in the unsupervised loss in Eq. (C.27). To evaluate the learned representations on downstream tasks, we typically fix $h$ and train a linear classifier $\mathbf{W} \in \mathbb{R}^{m \times d}$ on the top of the encoder $h$. Therefore, the supervised loss of $h$ at at $(\mathbf{x}, y)$ is defined as:

$$L_{sup}(\mathbf{x}, y; h) := \inf_{\mathbf{W} \in \mathbb{R}^{m \times d}} L_{sup}(\mathbf{x}, y; \mathbf{W}h), \tag{C.29}$$

**Lemma 2** (Connection between GCL and Downstream Tasks [39]). *Given a input sample $\mathbf{v}$ with its positive sample $\mathbf{v}^+$ and the set of negative samples $\mathbf{V}^- = \{\mathbf{v}_1^-, \dots, \mathbf{v}_n^-\}$. $\mathcal{C}$ is the set of latent class with distribution $\eta$. The latent class of $\mathbf{v}^+$ is denoted as $c^+ \in \mathcal{C}$. Suppose any $\mathbf{v}^- \in \mathbf{V}^-$ has the latent class $c^- \in \mathcal{C}$. Then we have*

$$L_{un}(\mathbf{v}; h) \geq (1 - \tau) L_{sup}(\mathbf{v}, c^+; h) + \tau, \tag{C.30}$$

*where $\tau = \mathbb{E}_{c^+, c^- \sim \eta} \mathbf{1}\{c^+ = c^-\}$, which indicates the expectation that $c^+ = c^-$.*

*Proof.* Based on the definitions of $L_{un}(\mathbf{v}; h)$ and $L_{sup}(\mathbf{v}; h)$ in Eq. (C.27) and Eq. (C.29), we have

$$
\begin{aligned}
L_{un}(\mathbf{v}; h) &= \mathop{\mathbb{E}}_{c^+, c^- \sim \eta^2} \big[ \mathop{\mathbb{E}}_{\mathbf{v}^+ \sim \mathcal{D}_{c^+}, \mathbf{v}^- \sim \mathcal{D}_{c^-}} \ell(h(\mathbf{v})^\top (h(\mathbf{v}^+) - h(\mathbf{v}^-))) \big] \\
&\geq \mathop{\mathbb{E}}_{c^+, c^- \sim \eta^2} \ell(h(\mathbf{v})^\top ( \mathop{\mathbb{E}}_{\mathbf{v}^+ \sim \mathcal{D}_{c^+}} [h(\mathbf{v}^+)] - \mathop{\mathbb{E}}_{\mathbf{v}^- \sim \mathcal{D}_{c^-}} [h(\mathbf{v}^-)])) \\
&= (1 - \tau) \mathop{\mathbb{E}}_{c^+, c^- \sim \eta^2} [L_{sup}(\mathbf{v}, c^+; f) | c^+ \neq c^-] + \tau \\
&= (1 - \tau) L_{sup}(\mathbf{v}, c^+; h) + \tau.
\end{aligned}
\tag{C.31}
$$

This completes proof. $\qquad\square$

Note that the above bound is similar to Lemma 4.3 in [39]. By leveraging our Theorem 2, we can establish the connection between GCL and downstream tasks, and use this connection to prove the transferability of the certified robustness of GCL to downstream tasks.

**Theorem 3.** *Given a GNN encoder $h$ trained via GCL and an clean input $\mathbf{v}$. $\mathbf{v}^+$ and $\mathbf{V}^- = \{\mathbf{v}_1^-, \cdots, \mathbf{v}_n^-\}$ are the positive sample and the set of negative samples of $\mathbf{v}$, respectively. Let $c^+$ and $c_i^-$ denote the latent classes of $\mathbf{v}^+$ and $\mathbf{v}_i^-$, respectively. Suppose $f$ is the downstream classifier that classify a data point into one of the classes in $\mathcal{C}$. Then, we have*

$$
\mathbb{P}(f(h(\mathbf{v})) = c^+) > \max_{\mathbf{v}_i^- \in \mathbf{V}^-} \mathbb{P}(f(h(\mathbf{v})) = c_i^-)
\tag{C.32}
$$

*Proof.* Since $\mathbf{v}$ is a clean input, according to Eq. (C.4) and Sec. 4.1, the positive pair $(\mathbf{v}, \mathbf{v}^+)$ a pair of similar data that come from the same class distribution $\mathcal{D}_{c^+}$ and they have the following relationship:

$$
\mathbb{P}(\mathbf{v} \in \mathbb{B}(\mathbf{v}^+); h) > \max_i \mathbb{P}(\mathbf{v} \in \mathbb{B}(\mathbf{v}_i^-); h).
\tag{C.33}
$$

Then, based on Theorem 1, we know that $\mathbb{P}(\mathbf{v} \in \mathbb{B}(\mathbf{v}^+); h)$ is monotonically increasing as $s(h(\mathbf{v}), h(\mathbf{v}^+))$ increases. Therefore, we have

$$
s(h(\mathbf{v}), h(\mathbf{v}^+)) > \max_i s(h(\mathbf{v}), h(\mathbf{v}_i^-))
\tag{C.34}
$$

According to Eq. (C.34), we can obtain that:

$$
\sum_{i=1}^{n} [s(h(\mathbf{v}), h(\mathbf{v}^+)) - s(h(\mathbf{v}), h(\mathbf{v}_i^-))] > 0,
\tag{C.35}
$$

and the equivalent form of Eq. (C.35) is given as:

$$
(\tilde{s}(h(\mathbf{v}^+)) - \tilde{s}(h(\mathbf{v}^-)))^\top h(\mathbf{v}) > 0, \forall \mathbf{v}^- \in \{\mathbf{v}_1^-, \cdots, \mathbf{v}_n^-\},
\tag{C.36}
$$

where $\tilde{s}$ is the $l_2$-normalization operation on vector $\mathbf{v}$. Then, we can obtain:

$$
(\tilde{s}(h(\mathbf{v}^+)) - \tilde{s}(h(\mathbf{v}^-)))^\top \tilde{s}(h(\mathbf{v})) > 0, \forall \mathbf{v}^- \in \{\mathbf{v}_1^-, \cdots, \mathbf{v}_n^-\}.
\tag{C.37}
$$

To relate the robustness of GCL to that of downstream tasks, we select the negative samples whose latent class $c_i^-$ is different from $c^+$ and obtain the following relationship:

$$
\sum_{i=1}^{n} [(\tilde{s}(h(\mathbf{v}^+)) - \tilde{s}(h(\mathbf{v}^-)))^\top \tilde{s}(h(\mathbf{v})) | c^+ \neq c_i^-] > 0.
\tag{C.38}
$$

Substitute the left side of Eq. (C.38) into Eq (C.27), we have $L_{un}(\mathbf{v}; h) < 1$. According to Lemma 2, we know that:

$$
L_{sup}(\mathbf{v}, c^+; h) \leq \frac{L_{un}(\mathbf{v}; h) - \tau}{1 - \tau} < 1.
\tag{C.39}
$$

Hence, according to the definition of $L_{sup}$ in Eq. (C.29), we have:

$$
\mathbb{P}(f(h(\mathbf{v})) = c^+) > \mathbb{P}(f(h(\mathbf{v})) = c^-), \forall c^- \in \{c_1^-, \cdots, c_n^-\}
\tag{C.40}
$$

which means that the logit output of the positive latent class $c^+$ is always larger than any negative latent class $c_i^-$ for $\mathbf{v}$. Thus, we can rewrite Eq. (C.40) to Eq. (C.32), and conclude the proof. $\quad\square$

---

**Algorithm 1** The Training Algorithm of RES.

---

**Input:** $\mathcal{G} = (\mathcal{V}, \mathcal{E}, \mathbf{X})$.
**Output:** Trained GNN encoder $h_\theta$.
1: Randomly initialize $\theta$ for $h_\theta$;
2: **for** epoch=$1, 2, \ldots,$ **do**
3:      Generate two augmented graphs $\mathcal{G}_i$ and $\mathcal{G}_j$ by $q_i(\mathcal{G}) \sim \mathcal{T}$ and $q_j(\mathcal{G}) \sim \mathcal{T}$;
4:      Inject randomized edgedrop noise $\epsilon$ to $\mathcal{G}_i$;
5:      Obtain node or graph representation $\mathbf{Z}_i$ and $\mathbf{Z}_j$ from $\mathcal{G}_i$ and $\mathcal{G}_j$ by using $h_\theta$;
6:      Update $\theta$ by applying gradient descent to minimize Eq. (1).
7: **end for**
8: **return** $h_\theta$;

---

By applying Theorem 3, we can demonstrate that given $\mathbf{v}$'s perturbed version $\mathbf{v}' = \mathbf{v} \oplus \delta$, where $||\delta||_0 \leq k$, if $\mathbf{v}$ and $\mathbf{v}'$ satisfy Eq. (8) and Eq. (9) in Theorem 2, we have

$$\mathbb{P}(f(h(\mathbf{v}' \oplus \epsilon)) = c^+) > \max_{\mathbf{v}_i^- \in \mathbf{V}^-} \mathbb{P}(f(h(\mathbf{v}' \oplus \epsilon)) = c_i^-), \ \forall \, ||\delta||_0 \leq k, \tag{C.41}$$

which implies provable $l_0^k$-certified robustness retention of $h$ at $(\mathbf{v}, c^+)$ in downstream tasks. The proof is completed.

## D    Training Algorithms

We summarize the training method of Sec. 5.1 for training smoothed GNN encoders in Algorithm 1. Specifically, at each training epoch, we first generate two augmented graphs $\mathcal{G}_i$ and $\mathcal{G}_j$ via $q_i(\mathcal{G})$ and $q_j(\mathcal{G})$, respectively, where $q_i(\mathcal{G})$ and $q_j(\mathcal{G})$ are two graph augmentations sampled from an augmentation pool $\mathcal{T}$. The graph augmentation includes edge perturbation, feature masking, node dropping, etc. (line 3). Then we inject randomized edgedrop noise $\epsilon$ to one of the augmented graphs $\mathcal{G}_i$ (line 4). From line 5 to line 10, we train the GNN encoder $h_\theta$ through GCL by maximizing the agreement of representations in these two views. In detail, we apply $h_\theta$ to obtain node or graph representations $\mathbf{Z}_i$ and $\mathbf{Z}_j$ from $\mathcal{G}_i$ and $\mathcal{G}_j$, respectively (line 5), then we do gradient descent on $\theta$ based on Eq. (1).

## E    Discussions

### E.1    Difference between RES and Edge-dropping Augmentations in [9] and [58]

Our RES is inherently different from the random edge-dropping augmentation in GraphCL [9] and the learnable edge-dropping augmentation in ADGCL [58]: (**i**) Random edge-dropping is an augmentation method to generate different augmented views and maximize the agreement between views, and the learnable edge-dropping [58] is also an augmentation method to enhance downstream task performance. However, RES is devised from the robustness perspective, providing certifiable robustness and enhancing the robustness of any GCL method. (**ii**) While random edge-dropping and learnable edge-dropping are only applied to augment graphs for GCL, RES extends beyond this. Following the generation of two augmented views as shown in Sec. 5.1, RES injects randomized edgedrop noise into one augmented view during GCL training. Then, it performs randomized edgedrop smoothing in the inference phase through Monte Carlo, as shown in Sec. 5.2. Specifically, for inference using RES, $\mu$ samples of $h(\mathbf{v} \oplus \epsilon)$ are drawn by injecting randomized edge-drop noise $\epsilon$ to $\mathbf{v}$ $\mu$ times. The final prediction is from Monte Carlo, selecting the $\mu$ predictions with the highest frequency in $\mu$ samples.

To demonstrate the effectiveness of RES, we further compare ADGCL with RES-GraphCL on MUTAG and PROTEINS. We also add RES-ADGCL into comparisons. More details are shown in Appendix. I.4.

### E.2    Additional Details of Concatenation Vector

The concatenation vector $\mathbf{v}$ is a vector to depict the structure of the node/graph for learning representations. For node-level tasks, it represents the connection status of any pair of nodes in the K-hop

subgraph of the node $v$. For graph-level tasks, it represents the connection status of any pair of nodes in the graph $\mathcal{G}$. To construct such a vector, we select the upper triangular part of the adjacency matrix of the K-hop subgraph of $v$ or the graph $\mathcal{G}$ and flatten it into the vector, where each item in this vector can denote the connection status of any pair of nodes in the K-hop subgraph of the node $v$ or the graph $\mathcal{G}$.

The motivation for using this notation is that since we focus on perturbations on the graph structure $\mathbf{A}$ in this paper, we treat the feature vector of $v$ as a constant and use the adjacency matrix of the K-hop subgraph of the node or the adjacency matrix of the graph to represent the structure of the node or graph. For simplicity and clarity, given a GNN encoder $h$ and the concatenation vector $\mathbf{v}$ of the node $v$ or the graph $\mathcal{G}$ as above, we then omit the node feature matrix $\mathbf{X}$ and simply write the node $v$'s representation $h_v(A, X)$ and the graph $\mathcal{G}$'s representation $h(\mathcal{G})$ as $h(\mathbf{v})$. Therefore, we use a unified notation $\mathbf{v}$ to denote the node $v$ or the graph $\mathcal{G}$, and further facilitate our theoretical derivations.

### E.3 Definition of Well-trained GNN encoders

The well-trained GNN encoder is defined as an encoder that can extract meaningful and discriminative representations by mapping the positive pairs closer in the latent space while pushing dissimilar samples away.

To evaluate whether a GNN encoder $h$ is well trained or not mathematically, we introduce criteria based on the similarity between node/graph representations in the latent space. For each positive pair $(\mathbf{v}, \mathbf{v}^+)$ with its negative samples $\mathbf{V}^- = \{\mathbf{v}_1, \cdots, \mathbf{v}_n\}$, we clarify that $h$ is well-trained at $(\mathbf{v}, \mathbf{v}^+)$ if the following inequality is satisfied:

$$s(h(\mathbf{v}), h(\mathbf{v}^+)) > \max_{\mathbf{v}^- \in \mathbf{V}^-} s(h(\mathbf{v}), h(\mathbf{v}^-)), \tag{E.42}$$

where $s(\cdot, \cdot)$ is a cosine similarity function. This implies that $h$ can effectively discriminate $\mathbf{v}$ from all its negative samples in $\mathbf{V}^-$ and learn the meaningful representations for $\mathbf{v}$ in the latent space. Therefore, based on Eq. (E.42), we can further extend the criteria for certifying robustness in GCL, which is shown in Definition 2.

### E.4 Rationale behind Setting High Values for $\beta$

The proposed robust encoder training method in Sec. 5.1 improves the model utility of GCL. Even setting a large $\beta$ for RES, we can still obtain high robust accuracy on clean graphs, further leading to high certified accuracies.

Specifically, as shown in Sec. 6.1, certified accuracy denotes the fraction of correctly predicted test nodes/graphs whose certified perturbation size is not smaller than the given perturbation size. It implies that these certified robust samples should also be correctly predicted by RES in the clean datasets. However, as the reviewer said, introducing randomized edgedrop solely to test samples during the inference could potentially hurt downstream task performance and further negatively impact the certified robustness based on Eq.(8). Thus, we propose robust encoder training for RES in Sec. 5.1 by injecting randomized edgedrop noise into one augmented view during GCL . It ensures the samples with randomized edgedrop noises align in latent class with clean samples under the encoder, thereby mitigating the negative impacts of such noises and further benefiting the robustness and certification of RES.

## F Code

Our code is available at `https://github.com/ventr1c/RES-GCL`.

## G Additonal Details of Experiment Settings

### G.1 Dataset Statistics

For node classification, we conduct experiments on 4 public benchmark datasets: Cora, Pubmed [47], Amazon-Computers [48] and OGB-arxiv [49], Cora and Pubmed are small citation networks. Amazon-Computers is a network of goods represented as nodes, and edges between nodes represent

that the two goods are frequently bought together. OGB-arxiv is a large-scale citation network. For graph classification, we use 3 well-known dataset: MUTAG, PROTEINS [50] and OGB-molhiv [49]. MUTAG is a collection of nitroaromatic compounds. PROTEINS is a set of proteins that are classified as enzymes or non-enzymes. OGB-molhiv is a dataset contains molecules, which is adopted from MoleculeNet [59]. Among these datasets, for Cora and Pubmed, we evaluate the models on the public splits. Regarding to the other 5 datasets, Coauthor-Physics, OGB-arxiv, MUTAG, PROTEINS and OGB-molhiv, we instead randomly select $10\%$, $10\%$, and $80\%$ nodes/graphs for the training, validation and test, respectively. The statistics details of these datasets are summarized in Table 2.

Table 2: Dataset Statistics

| Datasets | #Graphs | #Avg. Nodes | #Avg. Edges | #Avg. Feature | #Classes |
|---|---|---|---|---|---|
| Cora | 1 | 2,708 | 5,429 | 1,443 | 7 |
| Pubmed | 1 | 19,717 | 44,338 | 500 | 3 |
| Coauthor-Physics | 1 | 34,493 | 495,924 | 8415 | 5 |
| OGB-arxiv | 1 | 169,343 | 1,166,243 | 128 | 40 |
| MUTAG | 188 | 17.9 | 39.6 | 7 | 2 |
| PROTEINS | 1,113 | 39.1 | 145.6 | 3 | 2 |
| OGB-molhiv | 41,127 | 25.5 | 27.5 | 9 | 2 |

## G.2 Attack Methods

One of our goals is to show RES is robust to various structural noises, we evaluate RES on 4 types of structural attacks in evasion setting, i.e., Random attack, Nettack [11], PRBCD [51], CLGA [16] for both node and graph classification. The procedure of the evasion attack against GCL in transductive node classification is shown in Algorithm 2. The details of these attacks are described following:

1. **Random Attack**: We randomly add some noisy edges to the graphs for node classification and graph classification, respectively, until the attack budget is satisfied. Specifically, we consider two kinds of attack settings for node classification, that is, global random attack and targeted random attack. For global random attack, which is used in Sec. 6.2, we randomly inject some fake edges (10% in our setting) into the whole graph. For targeted attack, we randomly connect some fake edges to the direct neighbors of target nodes.

2. **Nettack** [11]: It is a targeted attacks for node classification that manipulate the graph structure to mislead the prediction of target nodes.

3. **PRBCD** [51]: It is a scalable global attack for node classification that aims to decrease the overall accuracy of the graph.

4. **CLGA** [16]: It is an unsupervised gradient-based poisoning attack targeting graph contrastive learning for node classification. Since we focus on evasion attacks, we directly use the poisoned graph generated by CLGA in the downstream tasks to evaluate the performance and regard it as an evasion global attack.

Moreover, we further consider two graph injection attack methods, i.e., TDGIA [60] and AGIA [61], as baselines to demonstrate the robustness of RES. The details of experimental results are shown in Sec. I.3.

## G.3 Compared Methods

We select four state-of-the-art GCL methods and employ RES on them to train smoothed GNN encoders:

1. **GRACE** [30]: It is node-level GCL method which creates multiple augmented views by perturbing graph structure and masking node features. Then it encourages consistency between the same nodes in different views.

2. **BGRL** [52]:Insipred by BYOL [62], it is performs graph contrastive learning that does not require negative samples. Specifically, it applies two graph encoders (i.e., online and targeted encoders), and update them iteratively to make the predicted representations closer to the true representations for each node.

**Algorithm 2** The Evasion Attack Procedure against GCL in Transductive Node Classification

---

**Input:** Clean graph $\mathcal{G} = (\mathcal{V}, \mathcal{E}, \mathbf{X})$, GNN encoder $h_\theta$, target node $v$ with class label $c$, perturbation $\delta$, downstream classifier $f$.
**Output:** success (i.e., attack $v$ successfully or not)
 1: Train $h_\theta$ on $\mathcal{G}$ via GCL;
 2: Obtain perturbed node $v'$ by adding perturbation $\delta$ to $v$;
 3: Generate the representation of $v'$ as $h(v')$;
 4: **if** $f(h(v')) = c$ **then**
 5:    **return** success $\leftarrow$ **false**;
 6: **else**
 7:    **return** success $\leftarrow$ **true**;
 8: **end if**

---

3. **DGI** [31]: It is a state-of-the-art GCL method which adapted from Deep InfoMax [63] to maximize the mutual information between local and global features.

4. **GraphCL** [9]: It is the first work to study GCL at graph-level. Specifically, it constructs four types of graph augmentations and adapts SimCLR [55] to learn graph-level embeddings.

Moreover, we also compare RES-GCL with several representative and state-of-the-art graph representation learning methods and robust GCLs against structural noises:

1. **Node2Vec** [53]: It is a traditional unsupervised methods. Its key idea is to perform random walks on the graph to generate sequences of nodes that capture both the local and global structure of the graph.

2. **GAE** [54]: It is a representative unsupervised learning method which learns a low-dimensional representation of a graph by encoding its nodes and edges into a vector space, and then decoding this representation back into a graph structure. It is trained to minimize the reconstruction error between the original graph and the reconstructed graph.

3. **GCL-Jaccard**: It is implemented by removing dissimilar edges based on Jaccard similarity before and after the training phase, respectively, which is inspired from GCN-Jaccard [64].

4. **Ariel** [15]: It is a robust GCL method which uses an additional adversarial graph view in graph contrastive learning to improve the robustness. An information regularizer is also applied to stabilize its performance.

### G.4 Implementation details

A 2-layer GCN is employed as the backbone GNN encoder and a common used linear evaluation scheme [31] is adopt in the downstream tasks. More specifically, each GNN encoder is firstly trained via GCL method and then the resulting embeddings are used to train and test a $l_2$-regularized logistic regression classifier. GRACE is implemented based on the source code published by authors [1]. BGRL and DGI methods are implemented based on PyGCL library [65]. All hyperparameters of all methods are tuned based on the validation set for fair comparison. All models are trained on an A6000 GPU with 48G memory.

### G.5 Attack Settings

In this paper, we assume that the attacker can conduct attack in two settings: transductive node classification and inductive graph classification. These two settings are described below:

- **Transductive Setting:** In this setting, test instances (nodes/graphs) are visible during both training the GNN encoder and inference in downstream tasks. Specifically, we first train a GNN encoder $h$ via GCL on a clean dataset that includes test nodes to generate node representations. Then, an attacker adds perturbations to the test nodes, causing $h$ to produce poor representations that degrade performance on downstream tasks. For example, an attacker may attempt to manipulate a social network by injecting fake edges, which could affect the performance of

---

[1]https://github.com/CRIPAC-DIG/GRACE

a well-trained $h$ in tasks such as community detection, influential node identification, or link prediction.

- **Inductive Setting:** In this setting, test instances only appear in downstream tasks and are invisible during the training phase. This setting is similar to the transductive setting, but the test nodes/graphs are not seen during training. This scenario commonly arises in real-world applications such as new drug discovery, where an attacker may attempt to manipulate a new molecular graph in the test set to mislead the model, resulting in incorrect predictions in downstream tasks.

## H   Additional Results of the Performance of Certificates

In this section, we extend the experiments in Sec. 6.3 and present comprehensive results on the performance of robustness certificates. Our aim is to demonstrate that RES can effectively provide certifiable robustness for various GCL methods. We select GRACE, BGRL, DGI, and GraphCL as the target GCL methods and integrate them with RES. We perform experiments on Cora, Pubmed, Coauthor-Physics, and OGB-arxiv for node classification tasks, as well as MUTAG, PROTEINS, and OGB-molhiv for graph classification tasks. *Certified accuracy* is selected as the evaluation metric. Specifically, *certified accuracy* [46, 24] denotes the fraction of correctly predicted test nodes/graphs whose certified perturbation size is no smaller than the given perturbation size. The complete results are reported in Fig. 4 to Fig. 8.

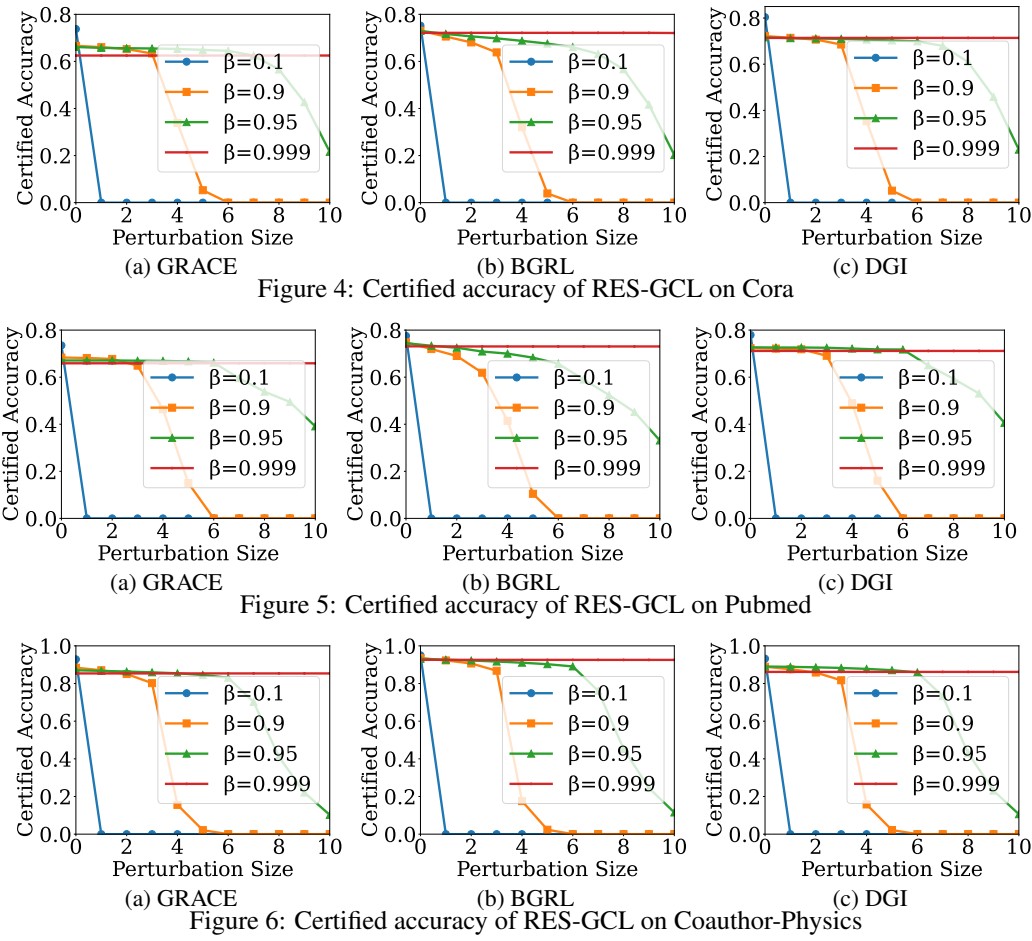

(a) GRACE        (b) BGRL        (c) DGI

Figure 4: Certified accuracy of RES-GCL on Cora

(a) GRACE        (b) BGRL        (c) DGI

Figure 5: Certified accuracy of RES-GCL on Pubmed

(a) GRACE        (b) BGRL        (c) DGI

Figure 6: Certified accuracy of RES-GCL on Coauthor-Physics

## I   Additional Results of the Performance of Robustness

In this section, we provide additional experimental results to further showcase the effectiveness of the RES in enhancing the robustness of GCL against various adversarial attacks. More specifically,

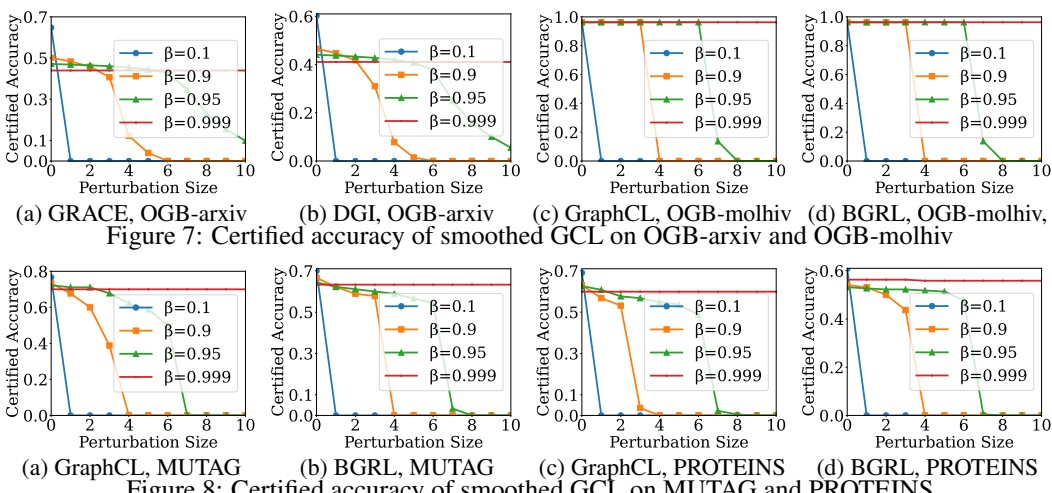

(a) GRACE, OGB-arxiv    (b) DGI, OGB-arxiv    (c) GraphCL, OGB-molhiv    (d) BGRL, OGB-molhiv,

Figure 7: Certified accuracy of smoothed GCL on OGB-arxiv and OGB-molhiv

(a) GraphCL, MUTAG    (b) BGRL, MUTAG    (c) GraphCL, PROTEINS    (d) BGRL, PROTEINS

Figure 8: Certified accuracy of smoothed GCL on MUTAG and PROTEINS

in Sec. I.1, we show the comparison results of RES with the baselines on graph classification. In Sec. I.2, we conduct experiments to demonstrate GCL with RES is resistant to different levels of structural noises. In Sec. I.3, we present the comparison results of three target GCL methods (i.e., GRACE, BGRL, DGI) against three types of structural attacks (i.e., Random, CLGA and PRBCD) on node classification. In Sec. I.4, we present additional experimental results of two advanced GCL methods (i.e., ADGCL [58] and RGCL [66]) on graph classification.

## I.1    Robust Performance on Graph Classification

In this subsection, we conduct experiments on graph classification to demonstrate the effectiveness of our method in this downstream task. Due to the limited availability of open-source attack methods specifically designed for graph classification, we utilize random attacks as the attack method. However, we believe that our method is robust against other attack methods as well. Specifically, we select GraphCL as the target GCL methods. The perturbation rate of random attack is 0.1. The smoothed version of GraphCL is denoted as RES-GraphCL. The results on MUTAG and PROTEINS are given in Fig. 9. From the figure, we observe: **(i)** When no attack is applied to the raw graphs, RES-GraphCL achieves comparable performance to the baseline GraphCL. **(ii)** When attacks are conducted on the noisy graphs, RES-GraphCL consistently outperforms the baseline on both the MUTAG and PROTEINS datasets. This result demonstrates the effectiveness of our method in enhancing the robustness of GraphCL against adversarial attacks.

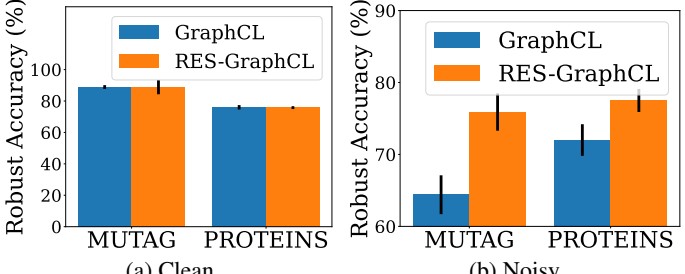

(a) Clean                 (b) Noisy

Figure 9: Robust accuracy of GraphCL for graph classification against random attack

## I.2    Robustness Under Different Noisy Levels.

To demonstrate the ability of our method to improve the robustness of Graph Contrastive Learning (GCL) against different levels of structural noise, we compare the robust accuracy of the GCL methods w/o applying our RES under evasion attacks for node classification. Specifically, we select GRACE, BGRL and DGI as target GCL methods. We set $(1 - \beta) = 0.05$ and $\mu = 50$. We consider two targeted attack method, random attack and Nettack [11] to conduct targeted attacks. The attack budget is set from 0 to 5. We randomly select 15% of the test nodes as the target nodes to compute the robust accuracy. The results on the Cora and Pubmed datasets are reported in Fig. 10 and Fig. 11. We observe: **(i)** The robust accuracies of the baseline methods exhibit a significant drop as the

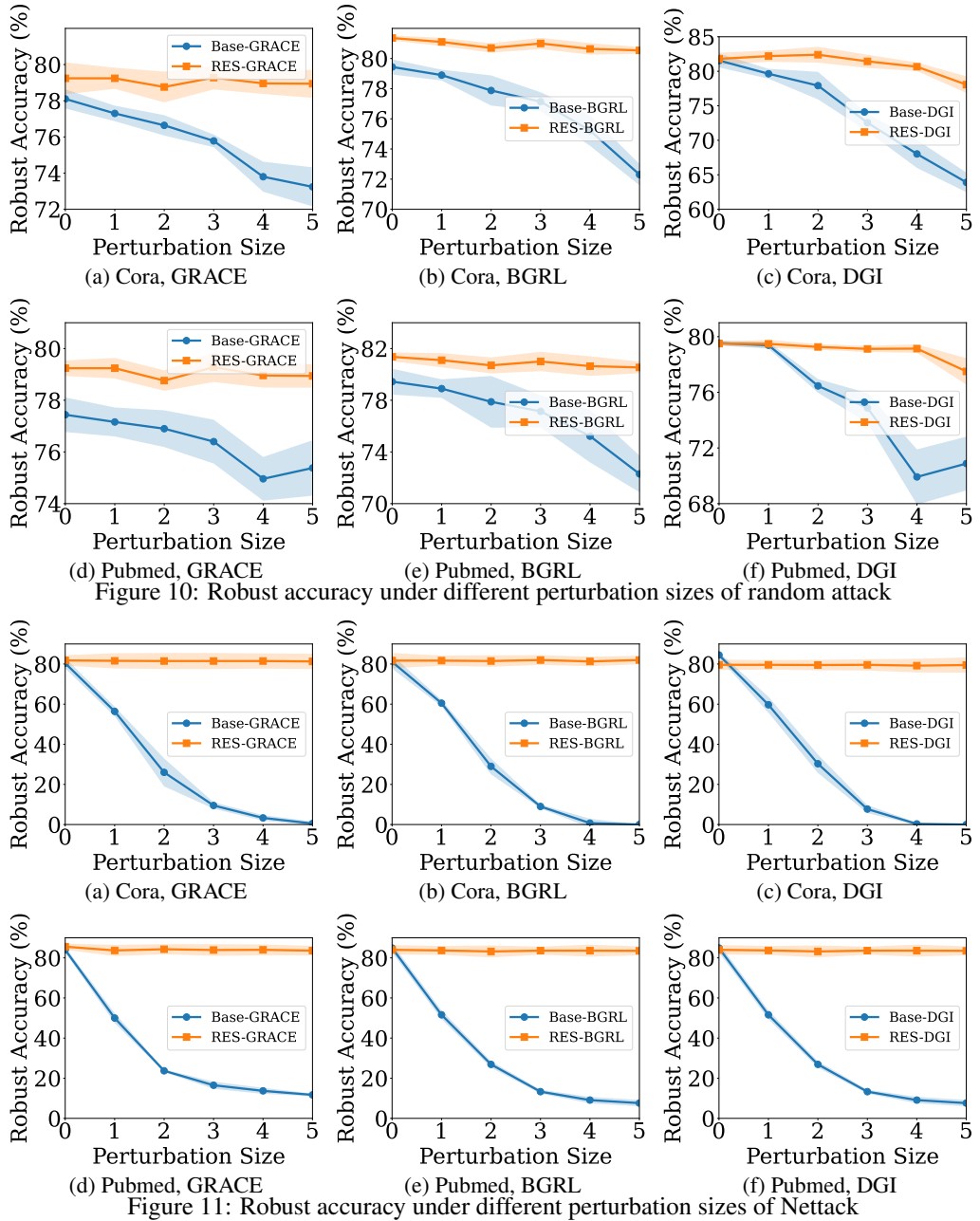

Figure 10: Robust accuracy under different perturbation sizes of random attack

Figure 11: Robust accuracy under different perturbation sizes of Nettack

perturbation sizes increase, which is expected. In contrast, the performances of RES are much more stable and consistently outperform the baselines. This demonstrates the robustness of GCL with RES against various levels of structural noise. **(ii)** The high robust accuracies of the three GCL models demonstrate that our RES is effectively applicable to various GCL method.

## I.3  Additonal Results of Robust Performance on Node Classification

In this subsection, we provide additional experimental results from Section 6.2, focusing on node classification. We select GRACE, BGRL, and DGI as the target GCL methods and evaluate their performance on four types of graphs: raw graphs, random attack perturbed graphs, CLGA perturbed graphs, and PRBCD perturbed graphs. The perturbation ratio is set to 0.1. Table 3 presents the results of these experiments. The highlighted results denote the best performance for each pair of GCL and RES-GCL. Note that BGRL, implemented based on PyGCL [65], encounters out-of-memory (OOM) errors in our platform, and hence the results for BGRL on OGB-arxiv are left blank. From

the table, we observe that all three GCL methods, when combined with RES, achieve state-of-the-art performance. This demonstrates the effectiveness of our method in enhancing the robustness of various GCL methods. By incorporating RES, the GCL models are more resilient to adversarial attacks and exhibit improved performance across different types of perturbed graphs.

Moreover, we further consider two graph injection attack methods, i.e., TDGIA [60] and AGIA [61], as baselines to demonstrate the robustness of RES. Specifically, we select GRACE as the target GCL methods and evaluate them on three types of graphs: raw graphs, TDGIA perturbed graphs and AGIA perturbed graphs. We insert the same number of fake nodes as the target nodes. We set $(1 - \beta) = 0.1$ and $\mu = 50$. The comparison results on four datasets are shown in Table. 4. From the results, we observe that (**i**) RES-GRACE consistently outperforms the baselines across 4 datasets in defending graph injection attacks. (**ii**) Both TDGIA and AGIA are much more powerful than the structural attack methods in Table 3 against GCL.

Table 3: Robust accuracy results of GCL methods for node classification.

| Dataset | Graph | GRACE | RES-GRACE | BGRL | RES-BGRL | DGI | RES-DGI |
|---------|-------|-------|-----------|------|----------|-----|---------|
| Cora | Raw | 77.1±1.6 | **79.7±1.0** | 78.5±1.6 | **79.9±1.2** | 81.3±0.7 | **81.4±0.8** |
| | Random | 74.5±2.1 | **79.7±1.0** | 76.2±1.2 | **79.6±1.1** | 77.5±1.0 | **79.2±0.6** |
| | CLGA | 74.9±2.0 | **78.2±1.0** | 75.8±1.6 | **79.4±0.9** | 79.5±0.5 | **80.9±1.0** |
| | PRBCD | 75.8±2.5 | **78.5±1.7** | 76.4±0.6 | **79.5±0.8** | 79.6±1.4 | **80.9±1.1** |
| Pubmed | Raw | 79.5±2.9 | **79.5±1.2** | 79.9±1.4 | **81.5±0.6** | **80.1±0.9** | 80.0±0.8 |
| | Random | 75.0±1.0 | **78.2±0.9** | 74.0±1.0 | **81.0±0.7** | 76.7±0.7 | **78.8±0.6** |
| | CLGA | 76.6±2.5 | **78.3±1.1** | 77.9±0.3 | **81.6±0.2** | 79.6±0.6 | **80.0±1.2** |
| | PRBCD | 73.2±2.3 | **78.8±1.7** | 71.6±2.4 | **80.9±0.5** | 75.4±0.9 | **78.0±0.6** |
| Physics | Raw | 94.0±0.4 | **94.7±0.2** | 95.3±0.1 | **95.6±0.1** | 93.5±0.6 | **94.1±0.3** |
| | Random | 92.6±0.5 | **94.2±0.3** | 94.0±0.2 | **95.5±0.2** | 91.5±0.9 | **92.2±0.4** |
| | PRBCD | 89.2±0.6 | **94.1±0.2** | 92.3±0.2 | **95.4±0.2** | 88.8±0.5 | **90.0±0.4** |
| OGB-arxiv | Raw | 65.1±0.5 | **65.2±0.1** | - | - | **65.0±0.2** | 64.8±0.1 |
| | Random | 59.0±0.2 | **60.0±0.1** | - | - | 58.0±0.1 | **58.9±0.1** |
| | PRBCD | 55.7±0.4 | **58.3±0.4** | - | - | 56.9±0.9 | **57.2±0.3** |

Table 4: Robust accuracy results of GCL methods against injection attacks.

| Dataset | Graph | GRACE | RES-GRACE |
|---------|-------|-------|-----------|
| Cora | Raw | 77.1±1.6 | **79.7±1.0** |
| | TDGIA | 22.4±1.5 | **78.7±1.1** |
| | AGIA | 21.1±1.4 | **78.4±0.9** |
| Pubmed | Raw | 79.5±3.1 | **79.5±1.2** |
| | TDGIA | 50.6±1.2 | **77.0±0.4** |
| | AGIA | 50.2±1.6 | **77.5±1.0** |
| Physics | Raw | 94.0±0.4 | **94.7±0.2** |
| | TDGIA | 52.2±1.0 | **93.8±0.1** |
| | AGIA | 52.3±0.5 | **94.0±0.2** |
| OGB-arxiv | Raw | 65.1±0.5 | **65.1±0.1** |
| | TDGIA | 40.3±0.4 | **46.9±0.1** |
| | AGIA | 40.4±0.3 | **47.0±0.1** |

## I.4 Additional Results of RES for Advanced GCLs

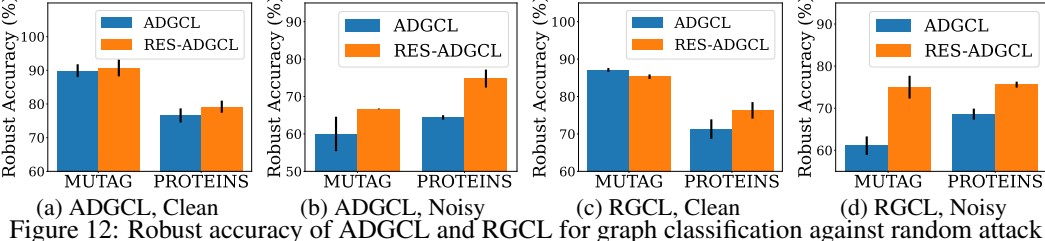

(a) ADGCL, Clean    (b) ADGCL, Noisy    (c) RGCL, Clean    (d) RGCL, Noisy

Figure 12: Robust accuracy of ADGCL and RGCL for graph classification against random attack

To further demonstrate the effectiveness of RES, we further add ADGCL [58] and RGCL [66] as baselines and implement RES-ADGCL and RES-RGCL. We set graph classification as the downstream task. The hyperparameter is tuned based on the performance of the validation set. We use random attack to get the noisy graphs and the perturbation rate is 0.1. Each experiment is conducted 5 times and the average results are reported. Comparison results on MUTAG and PROTEINS are

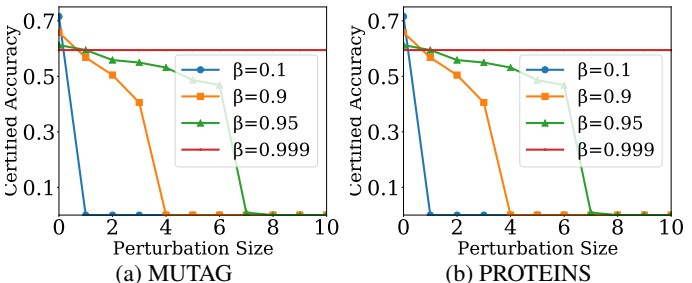

(a) MUTAG                    (b) PROTEINS

Figure 13: Certified accuracy of RGCL on MUTAG and PROTEINS

shown in Fig. 12. From this figure, we observe that all RES-GCL methods achieve comparable performances to the baselines on raw graphs and consistently outperform the baselines in the noisy graphs of two datasets, which validates the effectiveness of RES in any GCL model.

We also report the certified accuracy of RES-RGCL on the two datasets. The results are shown in Fig. 13. From the figure, we can observe there is a tradeoff between certified robustness and model utility, which is similar to that of Sec. 6.3.

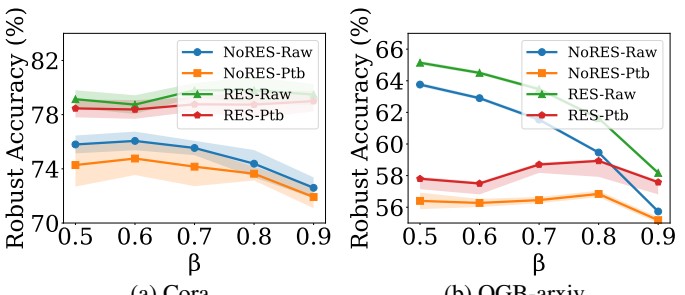

(a) Cora                    (b) OGB-arxiv

Figure 14: Ablation Studies of Training with RES on Cora and OGB-arxiv

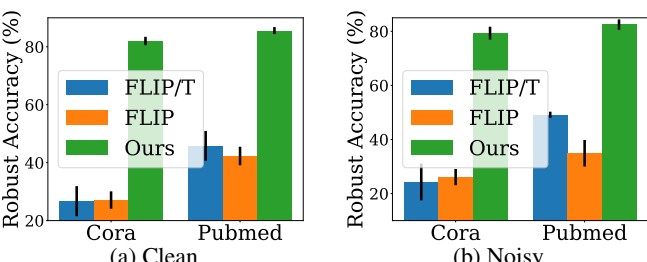

(a) Clean                    (b) Noisy

Figure 15: Ablation Studies of Comparisons RES with FLIP

# J    Additional Results of Ablation Studies

## J.1    Ablation Studies on Training with RES

In this subsection, we conduct ablation studies to investigate the effect of our training method for robust GCL. To demonstrate that our training method improves the robustness of GCL, we do not inject random edgedrop noise during the training phase and obtain a variant called NoRES. We select PRBCD as the attack method to generate noisy graphs, and set $\mu = 50$. We vary the value of $\beta$ as $\{0.5, 0.6, 0.7, 0.8, 0.9\}$ and compare the robust accuracy of RES and NoRES on both raw graphs and PRBCD-perturbed graphs. The results of the variant NoRES on raw and PRBCD-perturbed graphs are denoted as NoRES-Raw and NoRES-Ptb, respectively. We report the results on the Cora and OGB-arxiv datasets in Fig. 14. From the figure, we observe the following: **(i)** RES consistently outperforms NoRES on both raw and perturbed graphs in all settings, implying the effectiveness of our training method for robust GCL. **(ii)** The variance of RES is lower than that of NoRES. This is because we inject randomized edgedrop noise into the graphs during the training phase, helping the models better understand and generalize from the data with randomized edgedrop noise, thereby ensuring the robustness and utility of the models.

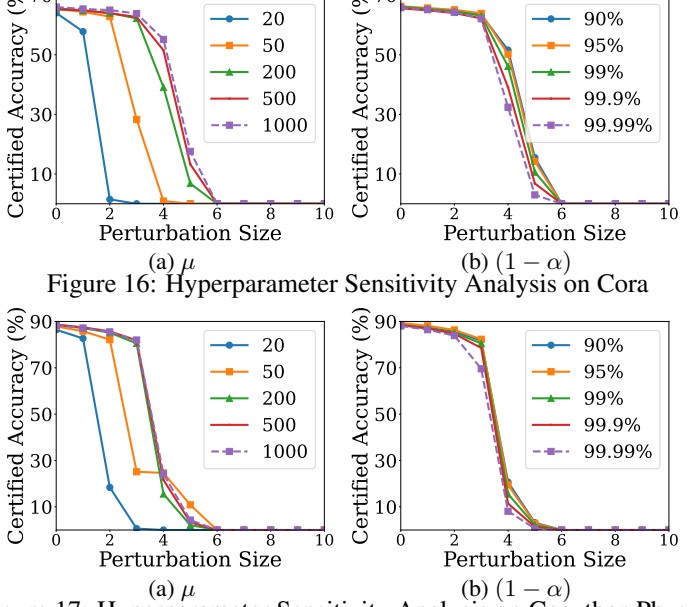

Figure 16: Hyperparameter Sensitivity Analysis on Cora

Figure 17: Hyperparameter Sensitivity Analysis on Coauthor-Physics

## J.2    Additional Results of Comparisons RES with FLIP

In this subsection, we present additional experimental results in Sec. 6.4 to further demonstrate that our RES method is significantly more effective in GCL compared to vanilla randomized smoothing [24]. We introduce two variants of our model, namely FLIP and FLIP/T, which replace the randomized edgedrop noise with binary random noise [24]. This noise flips the connection status within the graph with a probability of $\beta$. The key difference between FLIP and FLIP/T is that FLIP injects binary random noise into the graphs during both the training and inference phases, while FLIP/T only injects binary random noise during the inference phase. For our RES method, we set $\beta = 0.9$, and $\mu = 50$. For FLIP and FLIP/T, to ensure a fair comparison, we set $\mu = 50$, and vary $\beta$ over $\{0.1, 0.2, \cdots, 0.9\}$ and select the value that yields the best performance on the validation set of clean graphs. We select GRACE as the targeted GCL method and compare the robust accuracy of our RES with FLIP and FLIP/T on the clean and noisy graph under Nettack with an attack budget of 3. The average robust accuracy and standard deviation on Cora and Pubmed datasets are reported in Fig. 15. We observe: **(i)** RES consistently outperforms FLIP and FLIP/T on clean and noisy graphs of Cora and Pubmed datasets, further validating the effectiveness of RES in providing certified robustness for GCL. **(ii)** FLIP and FLIP/T exhibit comparable performance on all graphs, but significantly lower than RES. This finding confirms our analysis that vanilla randomized smoothing introduces numerous spurious/noisy edges to the graph, resulting in poor representation learning by the GNN encoder and compromising downstream task performance.

## K    Hyperparameter Sensitivity Analysis

We further investigate how hyperparameter $(1 - \alpha)$ and $\mu$ affect the performance of robustness certificates of our RES, where $(1 - \alpha)$ and $\mu$ control the confidence level and the number of Monte Carlo samples used to compute the certified accuracy. We vary the value of $(1 - \alpha)$ as $\{90\%, 95\%, 99\%, 99.9\%, 99.99\%\}$ and fix $\mu$ as 200, vary $\mu$ as $\{20, 50, 200, 500, 1000\}$ and fix $(1 - \alpha)$ as 99%, respectively. $\beta$ is set as 0.9. We report the certified accuracy of RES-GRACE on Cora and Coauthor-Physics dataset in Fig. 16 and Fig. 17. From the figures, we observe: **(i)** As $\mu$ increases, the certified accuracy curve becomes higher. The reason for this is that a larger value of $\mu$ makes the estimated probability bound $\underline{p_{\mathbf{v}+,h}}(\mathbf{v} \oplus \epsilon)$ and $\overline{p_{\mathbf{v}_i^-,h}}(\mathbf{v} \oplus \epsilon)$ in Eq. (8) tighter, resulting in a higher certified perturbation size of a test sample. **(ii)** As $(1 - \alpha)$ increases, the certified accuracy curve becomes slightly lower. This is because a higher confidence level leads to a looser estimation of $\underline{p_{\mathbf{v}+,h}}(\mathbf{v} \oplus \epsilon)$ and $\overline{p_{\mathbf{v}_i^-,h}}(\mathbf{v} \oplus \epsilon)$ in Eq. (8), meaning that fewer nodes satisfy Theorem 2 under the same perturbation size, resulting in a smaller certified perturbation size of a test sample.

# L    Limitations

In this paper, we propose a unified criteria to evaluate and certify the robustness of GCL. Our proposed approach, Randomized Edgedrop Smoothing (RES), injects randomized edgedrop noise into graphs to provide certified robustness for GCL on unlabeled data. Moreover, we design an effective training method for robust GCL by incorporating randomized edgedrop noise during the training phase. The theoretical analysis and extensive experiments show the effectiveness of our proposed RES.

**Limitation & future work**: Our current results are limited mainly to GCL while we believe it is also interesting to develop new techniques to other graph self-supervised methods e.g. generative and neighborhood prediction methods based on our framework, which we leave for immediate future work. We hope that this work could inspire future certifiably defense algorithms of adversarial attacks. Additionally, in this paper, we only focus on the graph-structured data. Thus, it is also interesting to investigate how to extend it to other domains, e.g., images and texts. Furthermore, in this paper, we utilize Monte Carlo algorithms to calculate robustness certificates for GCL, potentially increasing the computational demands. Therefore, it is also worthwhile to investigate methods to improve the efficiency of the robustness certification for GCL. Due to the nature of this work, there may not be any potential negative social impact that is easily predictable.

