# OpenReview forum: "Certifiably Robust Graph Contrastive Learning"
_NeurIPS.cc/2023/Conference — NeurIPS 2023 poster_

### Official Review · Reviewer_A65y · 2023-07-03

**Soundness:** 3 good
**Presentation:** 4 excellent
**Contribution:** 2 fair
**Rating:** 4
**Confidence:** 3

**Summary:**

Inspired by the certifiable robustness of graph contrastive learning (GCL) is still remain unexplored, the authors develop the first certifiably robust framework in GCL by proposing a unified criteria to evaluate and certify the robustness of GCL. Specifically, the authors introduce a novel technique, RES (Randomized Edgedrop Smoothing), to ensure certifiable robustness for any GCL model, and this certified robustness can be provably preserved in downstream tasks. Experiments on 7 real-world datasets show that the proposed RES-GRACE achieves state-of-the-art performance.

**Strengths:**

1. The presentation of the paper is very clear and the figures are easily digestible.
2. The numerical experimental results and visualization support the effectiveness of the proposed method.

**Weaknesses:**

1. In this paper, the authors conduct experiments for both node and graph classification tasks over regular and large-scale graphs. It is good. However, the choice of baselines seems somewhat insufficient. Can the authors compare with some recent GCL baselines? E.g., RGCL [1] and GLCC [2].

2. Can the authors provide some explanations why the performance of RES-backbone (e.g., RES-GRACE, RES-DGI) framework sometime is worse than the backbone on raw graph (however, their performance is always better than backbone under perturbation scenarios)?

3. From a clarity perspective, this work is, in my opinion, weakly motivated. It is not clear why the authors consider and start edgedrop smoothing to enhance the robustness of a GCL model.


[1] Li, Sihang, et al. "Let invariant rationale discovery inspire graph contrastive learning." International conference on machine learning. PMLR, 2022.

[2] Ju, Wei, et al. "Glcc: A general framework for graph-level clustering." Proceedings of the AAAI Conference on Artificial Intelligence. Vol. 37. No. 4. 2023.

**Questions:**

Please see comments and questions in Weaknesses.

**Limitations:**

The topic about certifiable robustness is quite interesting and is of high practical value. I believe this work has some potential things need to be addressed such as a better motivation and more details of comparisons in the experiments.

---

> ### Author Rebuttal · Authors · 2023-08-10
>
> We thank reviewer A65y for recognizing the novelties, the clear presentation, the valid technique details and extensively conducted experiments. The following is our point-to-point response to the reviewer’s concerns and comments:
>
> **Q1: Compare with some recent GCL baselines? E.g., RGCL [1] and GLCC [2].**
>
> We thank the reviewer for raising the related works. Note that since GLCC does not publish their source code for us, instead, we choose to only compare with RGCL, where we implement our RES on RGCL and obtain RES-RGCL. Specifically, we follow the same setting in the paper to compare the robust accuracy of RES-RGCL and RGCL. We set graph classification as the downstream task. The hyperparameter is tuned based on the performance of the validation set. We use random attack to get the noisy graphs and the perturbation rate is 0.1. Each experiment is conducted 5 times and the average results are reported. Comparison results in MUTAG and PROTEINS are shown in **Fig. 2 of the attached PDF file**.
>
> From the table, we can observe that (i) When no attack is applied to the raw graphs, RES-RGCL achieves comparable performance to the baseline RGCL. (ii) When attacks are conducted on the noisy graphs, RES-RGCL consistently outperforms the baseline on both the MUTAG and PROTEINS datasets. This result demonstrates the effectiveness of our method in enhancing the robustness of RGCL against adversarial attacks. The above observations are similar to that of GraphCL.
>
> We also report the certified accuracy of RES-RGCL on the two datasets. The results are shown in **Fig. 3 of the PDF file**. From the figure, we can observe the tradeoff between certified robustness and model utility, which is similar to that of Sec. 6.3.
>
> **Q2: Provide some explanations why the performance of RES-backbone (e.g., RES-GRACE, RES-DGI) framework sometime is worse than the backbone on the raw graph**
>
> Thanks for your careful reading. We would like to kindly clarify that the performance of RES-backbone framework is comparable to the baseline on raw graphs in these situations. For example, based on Table 3, the performance of RES-backbone is superior to the baselines on the raw graphs of all datasets except for Pubmed, OGB-arxiv.
>
> Specifically, RES-DGI achieves robust accuracy of 80.0±0.8 and 64.8±0.1 on the Pubmed and OGB-arxiv datasets, respectively. These results are comparable to the respective performance of DGI on the two datasets: 65.0±0.2 and 80.1±0.9. The difference in average robust accuracy between them is negligible and standard deviations of RES-DGI are also slightly lower than the baseline DGI, which validates our claim.
>
> **Q3: This work is weakly motivated. Why consider and start edgedrop smoothing to enhance the robustness of a GCL mode**
>
> Please see our following clarification for your concerns about our motivation.
>
> **(i)** Certified robustness aims to prove samples are robust to any perturbation in considered space. **To the best of our knowledge, there is no existing work studying the certified robustness of GCL.** Existing works [3,4,5] on the certifiable robustness of GNNs are designed for supervised settings, However, they generally analyze the worst-case attack on labeled nodes, which cannot be directly adapted to GCL due to the absence of labels.
>
> **(ii)** To address the above challenges, we propose **RES to derive certifiable robustness for GCL**. Our motivation is that injecting the randomized edgedrop noise $\epsilon$ to $\mathbf{v}’$ in the inference phase for multiple times will make each perturbed edge in the majority of these samples will be dropped, making these perturbed edges finally drop in the final decision. which makes certifying the robustness of GCL is feasible.
>
> **(iii)** However, applying RES to test samples in the inference solely may hurt performance in downstream tasks and the certified robustness based on Eq.(8). Thus, to address this issue and enhance RES’s robustness, we propose to train the robust GNN encoder by injecting randomized edgedrop noise to one augmented view and maximizing the agreement between two views via GCL to ensures the samples with randomized edgedrop noises have the same latent class as the clean samples. **This eliminates the negative impacts of randomized edgedrop noise, enhances model utility and robustness of GCL and further improves the robustness certification of GCL**.
>
> We again thank the reviewer for the time and effort in reviewing our paper. If you have any further concerns or questions, please do not hesitate to let us know. We will respond to them timely.
>
> [1] Let invariant rationale discovery inspire graph contrastive learning. ICML 2022.
>
> [2] Glcc: A general framework for graph-level clustering. AAAI 2023.
>
> [3] Certifiable robustness and robust training for graph convolutional networks. KDD 2019.
>
> [4] Efficient robustness certificates for discrete data: Sparsity-aware randomized smoothing for graphs, images and more. ICML 2020.
>
> [5] Certified robustness of graph neural networks against adversarial structural perturbation. SIGKDD 2021.

---

> > ### Comment · Reviewer_A65y · 2023-08-17
> > **Official Comment by Reviewer A65y**
> >
> > I appreciate the authors' responses and additional experiments. The authors have addressed my concerns. I will keep my score the same.

---

> > > ### Author Response · Authors · 2023-08-17
> > > **Response to Reviewer A65y**
> > >
> > > Thanks for your kind response. We are glad that your concerns are addressed. We sincerely hope that you can raise your rating. If any other concerns or questions are required to be addressed for raising scores, please let us know. We will respond to them timely.

---

### Official Review · Reviewer_vCmM · 2023-07-06

**Soundness:** 3 good
**Presentation:** 2 fair
**Contribution:** 2 fair
**Rating:** 5
**Confidence:** 3

**Summary:**

This paper introduces a certifiably robust graph contrastive learning method called RES (randomized Edgedrop Smoothing). This paper (1) first introduces the criteria of how to evaluate and certify robustness, (2) and then introduces RES to ensure certifiable robustness, (3) finally, proves that the certified robustness can be transferred to downstream tasks.

**Strengths:**

1. This paper is well motivated that current robust graph contrastive learning methods are not able to certify their robustness.
2. The theorem introduced in this paper looks interesting.
3. The experimental results on several benchmark datasets show that the proposed method could outperform baselines.

**Weaknesses:**

1. The motivation of using $\mathbf{v}^+$ rather than $\mathbf{v}$ in Eq. (3) is not quite clear. Specifically, in Eq. (3), why do you use $s(h(\mathbf{v}'), h(\mathbf{v}^+)) > s(h(\mathbf{v}'), h(\mathbf{v}^-))$ rather than $s(h(\mathbf{v}'), h(\mathbf{v})) > s(h(\mathbf{v}'), h(\mathbf{v}^-))$.
2. $v$ and $v^+$, and the term "positive sample" are a kind of confusing in section 4.1 (after line 194). I think the "positive sample" after line 194 has a different meaning of the "positive sample" in contrastive learning. It is highly suggested that the authors find a better way to present the idea here.
3. The claim in line 225-226 that "the majority of them will possess identical structural vectors, that is, $\mathbf{v}\oplus\mathbf{\epsilon}=\mathbf{v}'\oplus\mathbf{\epsilon}$" seems too strong. If this is the real case, can you show the statistics in some real datasets?
4. Random dropping edges is one of the basic graph augmentation techniques that have been introduced in GraphCL [1] and GraphMAE [2]. What's the difference between RES and random edge drop in GraphCL and GraphMAE?

[1] You, Yuning, et al. "Graph contrastive learning with augmentations." Advances in neural information processing systems 33 (2020): 5812-5823.
[2] Hou, Zhenyu, et al. "Graphmae: Self-supervised masked graph autoencoders." Proceedings of the 28th ACM SIGKDD Conference on Knowledge Discovery and Data Mining. 2022.

**Questions:**

1. In problem 1, what are $y$ and $y^\ast$?
2. What does "maximum of {$\mathbf{v}_1,\dots, \mathbf{v}_N$}" mean in Lemma 1? Can you show a concrete example?
3. There are many ways to define the probability shown in Eq. (5). For example, empirically, Gaussian is the most common and effective way. What are the advantages of your definition over a simple Gaussian?

---

> ### Author Rebuttal · Authors · 2023-08-10
>
> We thank reviewer vCmM for recognizing the novelties, the valid theoretical and technique details and extensive experiments of this work. The following is our point-to-point response to the reviewer’s concerns and comments:
>
> **W1: Why to use $\mathbf{v}^+$ instead of $\mathbf{v}$ in Eq. (3)**.
>
> Thanks for your question. We agree with the reviewer that $\mathbf{v}$ can be also used here because in Eq. (3), as $\mathbf{v}$ and $\mathbf{v}^+$ are the positive samples in GCL, which have the same latent class. Here are two reasons we use $\mathbf{v}^+$:
>
> **(i)** The use of $\mathbf{v}^+$ can distinguish it from $\mathbf{v}^-$, making our statement more clear.
>
> **(ii)** Since given a positive pair $(\mathbf{v}, \mathbf{v}^+)$, the goal of robustness certification of GCL is to make sure perturbed sample $\mathbf{v}’$ is always the positive sample of $\mathbf{v}^+$, as shown in Eq. (3) and Eq. (5). Replacing $\mathbf{v}$ as $\mathbf{v}’$ in Eq. (3) will cause the inconsistentness of notations in subsequent parts.
>
> **W2: "positive sample" after line 194 differs from the one in CL**
>
> Thanks for your question. We kindly clarify that both two “positive sample” share the same meaning. In GCL, the positive samples are generated from augmentation and they have the same latent classes based on the label-invariant augmentation intuition of GCL (see lines 129-130).
>
> For the “positive sample“ after line 194, it comes from Def. 2 about the certified robustness of GCL, where $(\mathbf{v}$, $\mathbf{v}^+)$ is a positive pair in GCL as shown in line 189. The meaning of Eq. (3) here is to check if $\mathbf{v}’$ under any perturbation within the specific budget is still most similar to the positive sample $\mathbf{v}^+$ of $\mathbf{v}$ than other $\mathbf{v}^+\_i$ in the latent space. In other words, it is to check if $\mathbf{v}’$ is still the positive sample of $\mathbf{v}^+$. Therefore, the two “positive sample” are the same.
>
>
> **W3: The claim that "the majority of them will possess identical structural vectors” seems too strong.**
>
> Thanks for your insightful question. We kindly clarify that given the perturbed sample $\mathbf{v}’=\mathbf{v}\oplus\delta$, after adding the randomized edgedrop noise $\epsilon$, the only difference between $\mathbf{v}’\oplus\epsilon$ and $\mathbf{v}\oplus\epsilon$ is in the part of fake edges in $\delta$. **Thus, the problem is reformulated to validate if the majority of the injected $\epsilon$ drop these fake edges.** Specifically, based on Sec. 5.2, RES will draw $\mu$ samples of $v’\oplus\epsilon$. Then use Monte Carlo to decide the final connection status of the edge by validating if the majority of the $\mu$ samples drop this edge. Thus, **all fake edges in most of the $\mu$ samples will be dropped if selecting high $\beta$**, making these fake edges finally dropped. which supports our claim.
>
> We further report the statistics on Pubmed to validate the correctness of the above statement. Specifically, we conduct Nettack with various perturbation sizes on test nodes. We conduct RES with various $\mu$ and $\beta$.  The frequency of test nodes satisfying $\mathbf{v}\oplus\epsilon=\mathbf{v}’\oplus\epsilon$ are shown in **Table 4 of the attached PDF file**. From the table, we observe that almost all test nodes possess $\mathbf{v}\oplus\epsilon=\mathbf{v}′\oplus\epsilon$, validating our claim.
>
> **W4: Difference between RES and random edgedrop in GraphCL and GraphMAE.**
>
> Thanks for your insightful question. We kindly clarify that our RES is inherently different from the random edgedrop in GraphCL and GraphMAE. Due to the word limits, please see the **Q1 of the global response** for our detailed responses.
>
> **Q1: What are $y$ and $y^\*$.**
>
> Thanks for pointing them out. $y^\*$ should be the latent class $c^\*$ of $\mathbf{v}$ in line 160 and $y$ should be $c$ in line 161. We apologize for the confusion and will revise them in the final version of our paper.
>
> **Q2: What does "maximum of $\\{\mathbf{v}_1,\cdots, \mathbf{v}_N\\}$" mean in Lemma 1?**
>
> Thanks for your question. “Maximum of $\\{\mathbf{v}_1,\cdots,\mathbf{v}_N\\}$” is the maximum of a sequence of i.i.d samples based on Lemma 1. In our theorem, it refers to $\\max\\{-D\_1,\cdots,-D\_{n}\\}$ in Eq.(C.9), where $D\_i=(1-s(h(\mathbf{v^+,v^-\_i})))/2$. It denotes the negative of half of the minimum distance between representations of positive and negative samples $\mathbf{v}^+$ and $\mathbf{v}^-\_i$. We apologize for this confusion about this notation and will revise it in the final version of our paper.
>
>  **Q3: Advantages of Eq.(5) over a simple Gaussian.**
>
> Thanks for your insightful question. The advantage of the definition of Eq. (5) is: **We can use Eq.(5) to reformulate the problem of certifying robustness for GCL in problem 1 to the problem of comparing the distances between positive and negative samples $\mathbf{v}^+$, $\mathbf{v}^-$ with respect to $\mathbf{v}$ as shown in Eq.(3). This enables an easier obtainment of GCL’s certified robustness.**
>
> Specifically, based on Problem 1 in Sec. 3.3, we aim to develop a certifiably robust GCL for encoder $h$ such that the probability of $h(\mathbf{v}’)$ in the latent class $c^*$ is always the largest over any other $c\neq c^*$. However, quantitatively calculating such a probability is indeed challenging in GCL. Based on Theorem 1 and its proofs in lines 630-648, Eq. (5) build a connection between the probability of $\mathbf{v}$ being the positive sample of $\mathbf{v}^+$ and the distance between them in the latent space. And the probability aligns with that of $\mathbf{v}’$ in latent class $c^*$ due to label-invariant augmentation intuition in line 129.
> Then, we can RES to derive certified robustness based on Eq. (5).
>
> [1] Graph contrastive learning with augmentations. NeurIPS 2020.
>
> [2] Adversarial graph augmentation to improve graph contrastive learning. NeurIPS 2021.
>
> [3] Label-invariant augmentation for semi-supervised graph classification. NeurIPS 2022.

---

> > ### Comment · Reviewer_vCmM · 2023-08-16
> >
> > Thanks for the response. I've read the rebuttal and I'd like to slightly raise my score.

---

> > > ### Author Response · Authors · 2023-08-16
> > > **Response to Reviewer vCmM**
> > >
> > > Thanks for your kind response and approval. We sincerely appreciate your time and efforts in improving our paper. If you have any further concerns or questions, please do not hesitate to let us know. We will respond to them timely.

---

### Official Review · Reviewer_F8a7 · 2023-07-06

**Soundness:** 3 good
**Presentation:** 3 good
**Contribution:** 2 fair
**Rating:** 5
**Confidence:** 3

**Summary:**

This study represents the first exploration of certifiable robustness in Graph Contrastive Learning (GCL). The authors propose a unified definition of robustness in GCL, addressing the existing ambiguity in quantifying its resilience to perturbations. The introduction of the Randomized Edgedrop Smoothing method is an interesting approach that applies randomized edgedrop noise into graphs to offer certifiable robustness on unlabeled data while minimizing the insertion of unnecessary edges. Theoretical analyses provided affirm the robust performance of their encoder in subsequent tasks. Extensive empirical experiments on various real-world datasets have been conducted, suggesting that the proposed method enhances the robustness of GCL models and provides certifiable robustness.

**Strengths:**

1. The authors explore a novel area of certifying the robustness of Graph Convolutional Learning (GCL) against various perturbations. They introduce a comprehensive definition for robustness in GCL and design an innovative framework, Randomized Edgedrop Smoothing (RES), to validate this robustness.
2. A theoretical analysis provided in the study affirms that the representations learnt by their robust encoder can deliver a provable robust performance in subsequent tasks.
3. The authors have developed an effective training strategy for robust GCL, integrating the application of randomized edgedrop noise.
4. Comprehensive empirical evaluations demonstrate that their method can deliver certifiable robustness for downstream tasks.


**Weaknesses:**

1. The selection of adversarial attack strategies in this study seems somewhat underwhelming given the marginal difference between clean accuracy and attack accuracy. It would be beneficial to employ more aggressive attack methods to genuinely assess the robustness of your proposed framework.
2. From what is shown in Table 3, the RES method appears to result in only slight improvements. Providing results from a broader range of graoh attack methods might lend more robust evidence to support your claims.
3. The paper seems to be missing a comparative analysis between the proposed Randomized Edgedrop Smoothing  method and existing edge drop augmentation techniques in Graph Contrastive Learning (GCL). Including such an analysis could provide readers with a more concrete understanding of the advantages and potential improvements brought about by the RES method.

**Questions:**

1. Could you clarify the operation of the evasion attack in the context of transductive node classification within Graph Contrastive Learning (GCL)?   In a scenario where structural perturbations are incorporated into the test set, however, it's worth noting that the GCL encoder is already trained based on the entire graph. So, in this transductive learning setting, what would be the practical significance of the evasion attack? Could you perhaps provide a schematic or flowchart, similar to Algorithm 1 in your paper, that visually represents the process of the attack method in GCL? This could facilitate a better understanding of its operation and mechanics, thus enriching the comprehension of your methodology.
2. I recommend considering stronger graph injection attacks for evasion scenarios, such as those presented in [1].
3. Could you provide a comparison between your proposed Randomized Edgedrop Smoothing (RES) method and the learnable edge-dropping augmentations described in [2]? I am interested in both theoretical distinctions and empirical differences based on experimental results.
4. Could you elucidate the distinction between your proposed Randomized Edgedrop Smoothing (RES) method and the conventional edge perturbation for augmentation as outlined in [3]? Understanding this difference could clarify the unique value proposition of your approach.
5. The definition of the "concatenation vector v" as given in line 106 seems somewhat ambiguous, as it appears to include the connected edge of node v. However, equation 5 also features h(v), where v presumably represents node features. Could you provide further clarification regarding the nature and role of the vector v?
6. In line 198, the space B is defined, but its details remain unclear. Could you provide a more comprehensive explanation of the space B, including its constituent elements? Is it meant to represent a probability space?

[1]. Chen Y, Yang H, Zhang Y, et al. Understanding and improving graph injection attack by promoting unnoticeability[J]. arXiv preprint arXiv:2202.08057, 2022.
[2]. Suresh S, Li P, Hao C, et al. Adversarial graph augmentation to improve graph contrastive learning[J]. Advances in Neural Information Processing Systems, 2021, 34: 15920-15933.
[3]. You Y, Chen T, Sui Y, et al. Graph contrastive learning with augmentations[J]. Advances in neural information processing systems, 2020, 33: 5812-5823.

---

> ### Author Rebuttal · Authors · 2023-08-10
>
> We thank reviewer F8a7 for recognizing the novelties, the valid theoretical and technique details and extensive experiments of this work. The following is our point-to-point response to the reviewer’s concerns and comments:
>
> **Q1: Clarify the operation of the evasion attack in the context of transductive node classification within GCL**
>
> Thanks for your thoughtful questions. Please find our answers below:
>
> **Evasion attack process.** In the context of transductive node classification within GCL, the target nodes of the evasion attack are visible during encoder training and inference. The objective is to perturb these target nodes, causing the GNN encoder $h$, to generate poor representations that degrade the nodes' performances on downstream tasks. The procedure of evasion attacks is summarized in **Fig. 1 of the attached PDF file.** Specifically, $h$ is first trained via GCL on the raw graph. Then perturbations are added to target node $v$, yielding a perturbed node $v{‘}$. After that, we generate the representations of $v{‘}$ via $h$ and evaluate its performance on the downstream tasks.
>
> **Practical significance.** In real-world scenarios like social networks and financial systems, pre-training a GNN encoder on untampered graphs is a common practice. Take social networks as an example: a company could pre-training a GNN encoder on the raw social network graph, and enable downstream task execution via an API. Consequently, user representations are periodically derived from the encoder for these tasks, involving users present during the encoder's training. Yet, malicious attackers might manipulate graph structure by injecting spurious edges, leading to mispredictions in downstream tasks and raising severe trustworthy concerns. Hence, ensuring the model's robustness against evasion attacks is also an important concern.
>
> **Q2: Compare with some stronger graph injection attacks in [1]**
>
> Thanks for the suggestions. We add two SOTA graph injection attack methods, i.e., TDGIA and AGIA [1]. We insert the same number of fake nodes as the target nodes. We compare the performance of GRACE and RES-GRACE. For RES-GRACE, we set $\mu=50$ and $\beta=0.9$. The comparison results on 4 datasets are shown in **Table 2 of the PDF file**. From the results, we observe that RES-GRACE consistently outperforms the baselines across 4 datasets in defending graph injection attacks.
>
> **Q3: Compare RES with the learnable edge-dropping augmentations in [2]**
>
> Thanks for sharing the great work [2]. We would like to kindly clarify that RES is inherently different from [2]:
>
> **(i)** Our work has different objectives as our method utilizes RES to **achieve certificate robustness for GCL**, while [2] is a learnable edge dropping augmentation to enhance downstream task performance.
>
> **(ii) In the inference phase, RES is deployed to guarantee certified robustness. Ans RES is further added to one augmented view during GCL training for model utility and robustness.** However, [2] is only used to augment graphs during training.
>
> To demonstrate the effectiveness of RES, we compare ADGCL with GraphCL. We also add RES-ADGCL into comparisons. We generate noisy graphs with a 10% random attack. The comparison results on two graph datasets are shown in **Table 3 of the PDF file**. From the table, we observe that RES-GraphCL and RES-ADGCL both achieve comparable performances to the baselines on raw graphs and consistently outperform the baselines in the noisy graphs of two datasets, which validates the effectiveness of RES in any GCL model.
>
> **Q4: Compare RES and the random edge-dropping in [3]**
>
> Thanks for your insightful question. We would like to kindly clarify that our RES is inherently different from the random edge-dropping augmentation in [3]. Due to the word limits, please see the **Q1 of the global response** for our detailed responses.
>
> **Q5: The nature and role of the concatenation vector v**
>
> Thanks for your insightful question. We would like to clarify that the concatenation vector $\mathbf{v}$ here is a vector to depict the structure of the node/graph for learning representations. Due to the word limits, please see the **Q2 of the global response** for our detailed responses.
>
>
> **Q6: Explain the space $\mathbb{B}$**
>
> Thanks for your question. In our paper, given a sample $\mathbf{v}^+$ with the latent class $c^+$,
> **$\mathbb{B}(\mathbf{v}^+)$ is a space for $\mathbf{v}^+$, where each constituent element within it is the positive sample of $\mathbf{v}^+$**. In the context of GCL, as detailed in 129, the label-invariant augmentation intuition of GCL makes augmented positive views have consistent latent classes with the original ones. Therefore, **each sample within this space also has the same latent class $c^+$, and we can then define $\mathbb{B}$ in Eq. (4).**
>
> **Role:** By leveraging the space $\mathbb{B}$ in Theorem 1 and 2, **we establish a connection between the probability of $\mathbf{v}$ being a positive sample of $\mathbf{v}^+$ and their cosine similarity in the latent space**. This connection enables the certification of GCL's robustness and transfers such robustness to downstream tasks to solve our problem 1 in Sec. 3.3.
>
> [1] Understanding and improving graph injection attack by promoting unnoticeability. ICLR 2022.
>
> [2] Adversarial graph augmentation to improve graph contrastive learning. NeurIPS 2021.
>
> [3] Graph contrastive learning with augmentations. NeurIPS 2020.

---

> > ### Comment · Reviewer_F8a7 · 2023-08-17
> >
> > Thank you for the authors' response.
> >
> > Regarding Q1, are you suggesting that by using RES $v'$ still obtains a robust representation after $h$?
> >
> > For Q3 and Q4, I would like to gain a deeper understanding of the role of RES during both the training and inference phases. Which part contributes to the robustness? In line 373 of your paper, could you clarify what is meant by 'the number of removed structures'?"

---

> > > ### Author Response · Authors · 2023-08-17
> > > **Response to Reviewer F8a7**
> > >
> > > Thanks for your time and effort in reviewing our paper. Here is our point-to-point response to your further questions:
> > >
> > > **Q1: are you suggesting that by using RES, $\mathbf{v}’$ still obtains a robust representation after $h$.**
> > >
> > > Thanks for your insightful question. We kindly claim that given the perturbed sample $\mathbf{v}’=\mathbf{v}\oplus\delta$, where $||\delta||_0<k$, by using RES in the GCL training and inference phases, the representation of $\mathbf{v}’$ learned by our RES-based encoder $h$ can achieve certifiably robust performance in downstream tasks, that is, $h(\mathbf{v}’)$ has consistently correct prediction in downstream tasks under any perturbation within budget $k$.
> > >
> > > **Q3 and Q4: The role of RES during both the training and inference phases. Which part contributes to the robustness?**
> > >
> > > Thanks for your insightful question. We kindly claim that RES works both in the training and inference phases, which contributes to the robustness together.
> > >
> > > **The role of RES in the inference phase:** RES performs randomized edgedrop smoothing in the inference phase through Monte Carlo. Our motivation is that injecting the randomized edgedrop noise $\epsilon$ to $\mathbf{v}’$ in the inference phase for multiple times will make each perturbed edge in the majority of these samples will be dropped, making these perturbed edges finally drop in the final decision. which makes certifying the robustness of GCL feasible and promises $\mathbf{v}’$’s robust performance. Specifically, based on Sec. 5.2, RES will draw $\mu$ samples of $\mathbf{v}’\oplus\epsilon$. Then we use Monte Carlo to decide the final connection status of the edge by validating if the majority of the $\mu$ samples drop this edge. Thus, **all fake edges in most of the $\mu$ samples will be dropped if selecting high $\beta$**, making these fake edges finally dropped.
> > >
> > > **The role of RES in the training phase:** RES is proposed to train a robust encoder in the training phase to eliminate the negative impacts of randomized edgedrop noise in inference, and enhances model utility and robustness of GCL. Specifically, our motivation is that applying RES to test samples in the inference solely may hurt performance in downstream tasks and the certified robustness based on Eq.(8). Thus, we propose robust encoder training for RES in Sec. 5.1 by injecting randomized edgedrop noise into one augmented view during GCL. It ensures the samples with randomized edgedrop noises align in latent class with clean samples under the encoder, thereby mitigating the negative impacts of such noises and further benefiting the robustness and certification of RES.
> > >
> > > **Which part contributes to the robustness:** According to the claim above, RES works both in the training and inference phases, which contributes to the robustness together. In the inference phase, our randomized edgedrop smoothing can drop all fake edges if selecting a high $\beta$ to ensure certifiably robust performance under any perturbation within the specific attack budget. In the training phase, our robust encoder training method mitigates the negative impacts of randomized edgedrop noise in inference and enhances the model utility of GCL, further benefiting the robustness and certification of RES.
> > >
> > > **The meaning of 'the number of removed structures' in line 373.**
> > >
> > > Thanks for your question. The number of removed structures denotes the number of views/graphs where all edges are removed. In the training phase, we usually have two views for GCL training. In the inference phase, we usually have one test graph. Therefore, we use $i\in\\{0,1,2\\}$ and $j\in\\{0,1\\}$ to denote the number of removed structures in the training and testing phases to implement several variants of our model in ablation studies to understand how RES contributes to the robustness of GCL.
> > >
> > > We again thank the reviewer for the time and effort in reviewing our paper. If you have any further concerns or questions, please do not hesitate to let us know. We will respond to them timely.

---

### Official Review · Reviewer_ssR8 · 2023-07-09

**Soundness:** 3 good
**Presentation:** 3 good
**Contribution:** 3 good
**Rating:** 6
**Confidence:** 3

**Summary:**

This paper studies the problem of certifiable robustness against adversarial perturbations in Graph Contrastive learning (GCL). This is an interesting paper where theoretical and empirical results are provided. The goal is to provide provable guarantee of robustness in the face of a limited budget adversarial attack on graph structure, showing that (1) positive samples remain positive samples in GCL, and (2) the downstream node/graph classification output does not change either. To this end randomized edge-drop smoothing (RES) is proposed and analyzed theoretically and empirically.

**Strengths:**

The main advantage of this work is its certification capability, which eliminates dependence on empirical assessment relying on empirical attacks, whose parameter/algorithmic tuning would be questionable.

The proposed method is simple and intuitive, and the experimental results show relative improvements on SOTA as well.

**Weaknesses:**

The main disadvantage of the proposed work seems to be its reliance on the latent classes in the downstream task, which is considered to be node/graph classification. This still encompasses a large set of problems and is of interest, however, it does partially violate the claim that a general GCL certification is studied.

Also, the manuscript has many typos and grammatical errors that would benefit from careful proof-reading.



**Questions:**

1- In multiple occasions in the definitions and theorems, an assumption is made on the encoder h being a "well-trained" GNN encoder, however this term is not defined, and it is not clear what criteria should be met for this. Since this is repeated in multiple Theorems, its clarification is crucial.

2- Is the importance of certification of the GCL in 9 itself only theoretical, and a stepping stone to providing certification on the downstream node/graph classification? Or is it possible to also evaluate such certification? I cannot seem to find any such results in the experiments section.

3- In ablation study in 6.4, what was the optimal $\beta$ for the FLIP algorithm? The gap between FlIP and other methods (Baseline and Ours) seems too large, and it is interesting to know if the smaller \beta=0.1 turned out to be the best value, which would motivate having a finer grained search for a smaller value for this.

4- What is the rationale behind setting such high values for $\beta$ in experiments in Figure 2? Such high values indicated that almost all edges are dropped in the randomization, which means that no structural information is retained, which is counterintuitive given that such high *certified* accuracy can still be obtained.

**Limitations:**

Increase in computational needs for Monte Carlo estimates and its environmental impacts should be mentioned in the limitations section.

---

> ### Author Rebuttal · Authors · 2023-08-10
>
> We thank reviewer ssR8 for recognizing the novelties, the valid theoretical and technique details and extensive experiments of this work. The following is our point-to-point response to the reviewer’s concerns and comments:
>
> **W1: The proposed work seems to rely on the latent classes in the downstream task, which is considered to be node/graph classification.**
>
> Thanks for your questions. We acknowledge that our work is based on the latent class to formalize GCL and establish the connection between the certified robustness of GCL and of downstream tasks. Many of the existing works [1,2,3] also exploit latent class to theoretically analyze contrastive learning. Thanks for your good suggestions. We leave other downstream tasks (e.g. link prediction) as future work.
>
> **W2: Typos and grammatical errors.**
>
> Thanks for pointing them out. We will correct them in the final version of this paper.
>
> **Q1: Define a “well-trained” GNN encoder.**
>
> Thanks for your insightful question. The definition of a “well-trained”: is an encoder that can extract meaningful and discriminative representations by mapping the positive pairs closer in the latent space while pushing dissimilar samples away.
>
> **Q2: The importance of the certification of GCL**
>
> Thanks for your insightful question. We would like to kindly clarify that our certified GCL robustness has following contributions:
>
> **(i)** Our work is the first attempt to certify the robustness of GCL.
>
> **(ii)** We propose a novel framework RES to certify GCL’s robustness. This framework also improves the robustness of any GCL model.
>
> **(iii)** Extensive experimental results on 7 real-world datasets validate the effectiveness of RES in certifying robustness and enhancing the robustness of any GCL model in practice. Please see Table 1 and Table 3 of the original paper for the details.
>
> **Q3: The optimal $\beta$ for the FLIP algorithm.**
>
> Thanks for your insightful question. To find the optimal $\beta$ for FLIP, we follow the same setting with various beta in Sec. 6.4 (see lines 362-363). The robust accuracies of FLIP on both clean and noisy Cora and Pubmed graphs under Nettack with an attack budget of $3$ are shown in **Table 1 of the attached PDF file**.
>
> Our observations are: (i) FLIP achieves the best robust accuracy on the clean graphs when $\beta=0.1$, implying that $\beta=0.1$ is the optimal choice. (ii) FLIP generally demonstrates inferior robust accuracy on the clean graphs at $\beta=0.9$. Our interpretation is that a smaller $\beta$ in FLIP introduces fewer noisy edges to clean graphs, thereby preserving more structural information, beneficial for downstream tasks. However, based on the analysis in [4], a smaller $\beta$ leads to a smaller certified perturbation size, which implies a tradeoff between model utility and certifiable robustness within FLIP. Therefore, it is important to study how to select the best $\beta$ in FLIP to balance the tradeoff. We will leave it as future work.
>
> **Q4: Rationale behind setting high values for $\beta$ in experiments.**
>
> The rationale behind setting high $\beta$ is: The proposed robust encoder training method in Sec. 5.1 improves the model utility of GCL. Even setting a large $\beta$ for RES, we can still obtain high robust accuracy on clean graphs, further leading to high certified accuracies.
>
> Specifically, as shown in line 321, certified accuracy denotes the fraction of correctly predicted test nodes/graphs whose certified perturbation size is not smaller than the given perturbation size. It implies that these certified robust samples should also be correctly predicted by RES in the clean datasets. However, as the reviewer said, introducing randomized edgedrop solely to test samples during the inference could potentially hurt downstream task performance and further negatively impact the certified robustness based on Eq.(8). Thus, we propose robust encoder training for RES in Sec. 5.1 by injecting randomized edgedrop noise into one augmented view during GCL . **It ensures the samples with randomized edgedrop noises align in latent class with clean samples under the encoder, thereby mitigating the negative impacts of such noises and further benefiting the robustness and certification of RES.**
>
> We sincerely thank you again for your time and efforts in improving our paper. We will include the above discussion in the final version of our paper. If you have any further concerns or questions, please do not hesitate to let us know. We will respond to them timely.
>
> [1] A theoretical analysis of contrastive unsupervised representation learning. ICML 2019.
>
> [2] Do more negative samples necessarily hurt in contrastive learning? ICML 2022.
>
> [3] Robustness verification for contrastive learning. ICML 2022.
>
> [4] Certified robustness of graph neural networks against adversarial structural perturbation. KDD 2021.

---

> > ### Comment · Reviewer_ssR8 · 2023-08-21
> > **Response to authors**
> >
> > Thank you for responding to the previous questions. Most of my questions are answered, however I am still not convinced with the qualitative definition of "well-trained" as also in the above response, it still remains unclear what is meant mathematically (e.g. in terms of minimum desired classification accuracy, or such quantities.) If no such requirement is necessary for the GNN encoder accuracy for the theorems to hold, I'd recommend to drop the term "well-trained" in the theorems to avoid being qualitative and mathematically unclear.
> >
> > I will keep my score unchanged.

---

> > > ### Author Response · Authors · 2023-08-21
> > > **Response to Reviewer ssR8**
> > >
> > > Thanks for your kind response. Please see the following clarification for your question about the definition of the well-defined GNN encoder:
> > >
> > > To evaluate whether a GNN encoder $h$ is well trained or not mathematically, we introduce criteria based on the similarity between node/graph representations in the latent space. For each positive pair $(\mathbf{v}, \mathbf{v}^+)$ with its negative samples $\mathbf{V}^- = \\{\mathbf{v}\_1,\cdots,\mathbf{v}\_n\\}$, we clarify that $h$ is well-trained at $(\mathbf{v},\mathbf{v}^+)$ if the following inequality is satisfied:
> > > $$s(h(\mathbf{v}),h(\mathbf{v}^+))>\max_{\mathbf{v}^-\in\mathbf{V^{-}}}{s(h(\mathbf{v}),h(\mathbf{v}^-))},$$
> > > where $s(\cdot,\cdot)$ is a cosine similarity function. This implies that $h$ can effectively discriminate $\mathbf{v}$ from all its negative samples in $\mathbf{V}^-$ and learn the meaningful representations for $\mathbf{v}$ in the latent space.
> > >
> > > Moreover, in this paper, we further extend the criteria for certifying robustness in GCL, as defined by Eq. (3). Based on the clarification above, given a GNN encoder $h$ that is well-trained at $(\mathbf{v}, \mathbf{v^{+}})$, suppose that $\mathbf{v}′$ is a perturbed sample obtained by adding structural noise $\delta$ to $\mathbf{v}$ as described in line 192, where $||\delta||\_0\leq k$. We then clarify that $h$ is certifiably robust at ($\mathbf{v}, \mathbf{v^{+}}$) if the following inequality is hold:
> > > $$s(h(\mathbf{v}{'}),h(\mathbf{v}^+))>\max_{\mathbf{v}^-\in\mathbf{V^{-}}}{s(h(\mathbf{v}{'}),h(\mathbf{v}^-))},~\forall{\delta}:\|\delta\|_{0} \leq k,$$
> > > which indicates that for any perturbation within the attack budget $k$, the cosine similarity $s(h(\mathbf{v}{'}),h(\mathbf{v}^+))$ is consistently larger than $s(h(\mathbf{v}{'}),h(\mathbf{v}^-))$ for any $\mathbf{v}^-\in{\mathbf{V}^-}$.
> > >
> > > We sincerely thank the reviewer again for your time and efforts in improving our paper. We will include the above discussion in the final version of our paper. If you have any further concerns or questions, please do not hesitate to let us know. We will respond to them timely.

---

### Official Review · Reviewer_661C · 2023-07-31

**Soundness:** 3 good
**Presentation:** 3 good
**Contribution:** 3 good
**Rating:** 7
**Confidence:** 4

**Summary:**

This paper is the first one to dive into the certifiably robust Graph Contrastive Learning (GCL) and proposes a certifiably robust GCL framework. It defines the certified robustness of GCL and then proposes Randomized EdgeDrop Smoothing (RES), which randomly drops each edge of the input sample with a certain probability. Besides, it also presents a simple training method for robust GCL. The theoretical analyses and extensive experiments demonstrate its effectiveness.

**Strengths:**

This paper is the first one to dive into the certifiably robust Graph Contrastive Learning (GCL) and proposes a certifiably robust GCL framework. It defines the certified robustness of GCL and then proposes Randomized EdgeDrop Smoothing (RES), which randomly drops each edge of the input sample with a certain probability. Besides, it also presents a simple training method for robust GCL. The theoretical analyses and extensive experiments demonstrate its effectiveness. The paper is well-written.

**Weaknesses:**

I have some minor comments, as below.
1. In Line 106, the definition of " concatenation vector \mathbf{v}" is confusing. What is the meanings of "which captures the upper triangular part of the adjacency matrix for the K-hop subgraph of v". Does it mean the K-hop neighbors of node v?
2. The use of probability operator symbols is inconsistent. For example, Eq. 2, 6, 7 use "\mathbb{P}", while Eq. 5 use "Pr". It is better to harmonize the symbols used.


**Questions:**

Please refer to the weaknesses above.

**Limitations:**

I believe this work does not have potential negative societal impacts.

---

> ### Author Rebuttal · Authors · 2023-08-09
>
> We are grateful for your approval of the novelties, the valid theoretical and technique details and extensively conducted experiments of our work. We sincerely thank you for your time and thoughtful feedback. We will perform all the changes requested in the minor comments. Here, we hope to address the raised points.
>
> **Q1: What is the meaning of "which captures the upper triangular part of the adjacency matrix for the K-hop subgraph of v"**
>
> Thanks for your insightful question. We would like to clarify the concatenation vector $\mathbf{v}$ here is a vector to represent the connection status of any pair of nodes in the K-hop subgraph of the node $v$, which can depict the structure of the K-hop subgraph of the node $v$. To construct such a vector, we select the upper triangular part of the adjacency matrix of the K-hop subgraph of $v$ and flatten it into the vector, where each item in this vector can denote the connection status of any pair of nodes in the K-hop subgraph of node $v$. A similar setting also appeared in [1].
>
> **Q2: The use of probability operator symbols is inconsistent in Eq.(5).**
>
> Thanks for pointing this out. We will correct this typo in the final version of this paper.
>
> We sincerely thank you again for your time and efforts in reviewing our paper. We will include the above discussion and revision in the final version of our paper. If you have any further concerns or questions, please do not hesitate to let us know. We will respond to them timely.
>
> [1] Certified robustness of graph neural networks against adversarial structural perturbation. KDD 2021.

---

> ### Comment · Area_Chair_Lthc · 2023-08-18
>
> Hi reviewer: Please kindly respond or acknowledge authors' rebuttal.

---

### Author Rebuttal · Authors · 2023-08-10

We sincerely thank all the reviewers for their thoughtful comments and constructive suggestions, which significantly helped us strengthen our paper. We are encouraged to find that the reviewers appreciate the novelty of this work, the valid theoretical and technique details and extensively conducted experiments and clear presentation. We now provide our answers to these shared comments and report the required additional experimental results and algorithms in **the attached PDF files**.

**Q1: Difference between RES and random edgedrop in GraphCL.**

Our RES is inherently different from the random edge-dropping augmentation in GraphCL:

**(i)** Random edge-dropping is an augmentation method to generate different augmented views and maximize the agreement between views. However, RES is devised from the robustness perspective, providing certifiable robustness and enhancing the robustness of any GCL method.

**(ii)** While random edge-dropping is only applied to augment graphs for GCL, RES extends beyond this. Following the generation of two augmented views as shown in Sec. 5.1, RES injects randomized edgedrop noise into one augmented view during GCL training. Then, it performs randomized edgedrop smoothing in the inference phase through Monte Carlo, as shown in Sec. 5.2. Specifically, for inference using RES (based on lines 274-276), $\mu$ samples of $h(\mathbf{v}\oplus\epsilon)$ are drawn by injecting randomized edge-drop noise $\epsilon$ to $\mathbf{v}$ $\mu$ times. The final prediction is from Monte Carlo, selecting the $\mu$ predictions with the highest frequency in $\mu$ samples.

**Q2: About the concatenation vector v.**

The concatenation vector $\mathbf{v}$ is a vector to depict the structure of the node/graph for learning representations. **For node-level tasks, it represents the connection status of any pair of nodes in the K-hop subgraph of the node $v$. For graph-level tasks, it represents the connection status of any pair of nodes in the graph $\mathcal{g}$.** To construct such a vector, we select the upper triangular part of the adjacency matrix of the K-hop subgraph of $v$ or the graph $\mathcal{g}$ and flatten it into the vector, where each item in this vector can denote the connection status of any pair of nodes in the K-hop subgraph of the node $v$ or the graph $\mathcal{g}$.

**Motivation:** The motivation for using this notation is that since we focus on perturbations on the graph structure $\mathbf{A}$ in this paper, we treat the feature vector of $v$ as a constant and use the adjacency matrix of the K-hop subgraph of the node or the adjacency matrix of the graph to represent the structure of the node or graph. For simplicity and clarity, given a GNN encoder $h$ and the concatenation vector $\mathbf{v}$ of the node $v$ or the graph $\mathcal{g}$ as above, **we then omit the node feature matrix $\mathbf{X}$ and simply write the node $v$’s representation $h\_{v}(A,X)$ and the graph $\mathcal{g}$’s representation $h(\mathcal{g})$ as $h(\mathbf{v})$.** Therefore, we use a unified notation $\mathbf{v}$ to denote the node $v$ or the graph $\mathcal{g}$, and further facilitate our theoretical derivations.

---

### Decision · Program_Chairs · 2023-09-21

**Decision:**

Accept (poster)

**Comment:**

AC read through the reviews, rebuttals and comments, and browse the paper quickly. All reviewers eventually agreed that RES is novel, even though there were some doubts about its difference from randomized edge dropping. Rebuttals seems to address that satisfactorily.